# *CivRealm*: A Learning and Reasoning Odyssey in *Civilization* for Decision-Making Agents

**Siyuan Qi**[1*†]   **Shuo Chen**[1*]   **Yexin Li**[1*]   **Xiangyu Kong**[1*]   **Junqi Wang**[1*]
**Bangcheng Yang**[1]   **Pring Wong**[1]   **Yifan Zhong**[1,2]   **Xiaoyuan Zhang**[1,2]
**Zhaowei Zhang**[1,2]   **Nian Liu**[1,3]   **Yaodong Yang**[2,1]   **Song-Chun Zhu**[1,2]

[1]National Key Laboratory of General Artificial Intelligence, BIGAI [2]Peking University [3]BUPT

## Abstract

The generalization of decision-making agents encompasses two fundamental elements: learning from past experiences and reasoning in novel contexts. However, the predominant emphasis in most interactive environments is on learning, often at the expense of complexity in reasoning. In this paper, we introduce *CivRealm*, an environment inspired by the *Civilization* game. *Civilization*'s profound alignment with human society requires sophisticated **learning** and prior knowledge, while its ever-changing space and action space demand robust **reasoning** for generalization. Particularly, *CivRealm* sets up an imperfect-information general-sum game with a changing number of players; it presents a plethora of complex features, challenging the agent to deal with open-ended stochastic environments that require diplomacy and negotiation skills. Within *CivRealm*, we provide interfaces for two typical agent types: **tensor-based** agents that focus on learning, and **language-based** agents that emphasize reasoning. To catalyze further research, we present initial results for both paradigms. The canonical RL-based agents exhibit reasonable performance in mini-games, whereas both RL- and LLM-based agents struggle to make substantial progress in the full game. Overall, *CivRealm* stands as a unique learning and reasoning challenge for decision-making agents. The code is available at https://github.com/bigai-ai/civrealm.

## 1 Introduction

Human intelligence behaves very differently from contemporary AI. From the earliest use of stone tools, our ancestors spent 18,000,000 years progressing to the control of fire [21]. If we were to learn and explore in a manner similar to modern AI agents, it would take even longer to transition from the industrial age to the information age. This is due to the exponential growth of our toolkit and action space $\mathcal{A}$, where finding a meaningful development trajectory $\tau$ in the vast realm of possibilities would seem nearly impossible. Yet, human society not only thrives but also at an ever-accelerating pace. In fact, it merely took 247 years from the invention of the steam engine to the birth of the digital computer [19; 30] . This is truly remarkable from a decision-making perspective: it is our ability to **learn** from past experience and **reason** in novel contexts that defies the probabilistic expectations.

In this paper, we present *CivRealm*, an interactive environment designed to push the boundaries of decision-making agents' learning and reasoning. *CivRealm* is inspired by the iconic game Civilization, where each agent acts as a player to guide a civilization, mirroring the course of human history. Within this game, players are confronted with decisions including resource management, strategic planning, diplomacy, and warfare. The game rules profoundly align with the mechanics governing human society, and decision-making in the game span from long-term strategic vision (*e.g.*, development prioritization) to fine-grained tactical maneuvers (*e.g.*, unit control) (Figure 1).

As a multi-agent interactive environment, *CivRealm* distinguishes itself through unique features that demand robust reasoning capabilities to adapt to its ever-changing conditions (Table 1). It establishes an imperfect-information general-sum game where the number of players fluctuates during gameplay. Within this setting, agents must adeptly navigate open-ended and stochastic environments with rapidly expanding state and action spaces, driven by technological advancements and societal progress. Additionally, effective interaction and communication between agents necessitate diplomatic and negotiation skills, akin to challenges encountered in real-world decision-making.

---

[*]Equal contribution. [†] Corresponding author.

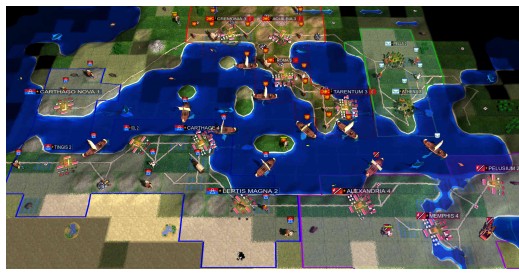 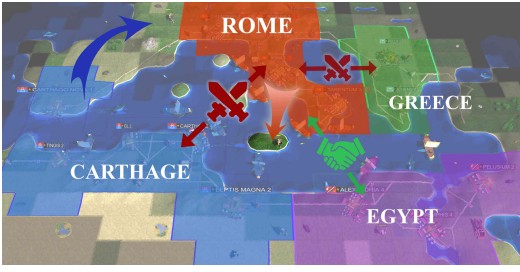

(a) Game interface.                                    (b) Examples of long-term strategies.

Figure 1: The gameplay of Civilization [73] requires deep reasoning, involving long-term strategic planning and fine-grained tactical controls. The figure depicts a hypothetical situation that resembles a historical scenario, with the Romans securing Sicily against Carthage while nurturing a friendly diplomatic relationship with a declining Egypt. This decision is made in a highly complex context: players need to consider various aspects of long-term developmental strategies like technology, military, and diplomacy in the given geographical and diplomatic context. They also engage in fine-grained control actions, *e.g.*, border exploration, vessel building, and road construction.

As interactive environments become increasingly complex [6; 76; 70; 2; 8; 15; 42; 43; 37; 44; 18], it is important to note that many lack the same degree of reasoning sophistication and dynamic complexity as *CivRealm*. One closely related environment is StarCraft [79]. However, being a real-time strategy game, it places a greater emphasis on quick tactical reactions, with a typical game concluding within 30 **minutes** for human players. In contrast, Civilization is a turn-based strategy game that simulates the development of human society, where a single game can span from several **hours** to even days to complete. This extended duration reflects the need for profound **reasoning** in this expanding decision space. On the other hand, statistics [83] show that human players learn to play the game reasonably well within 23 hours, which typically amounts to about **3-4 full games**. This presents the **learning** challenges for decision-making agents to match human speed.

In addition to the full game, *CivRealm* offers mini-games across three core aspects: development, battle, and diplomacy. The *development* mini-games challenge players to nurture their civilization's growth, including aspects like population, production, and economy. In *battle* mini-games, players need to master the control of multiple units against opposing forces. The *diplomacy* mini-games demand players to employ diplomatic actions (*e.g.*, trading), to foster their civilization's prosperity.

To stimulate future research, we provide both tensor and language APIs, accommodating two typical genres of decision-making agents: reinforcement learning (RL) and large language models (LLM) agents. We have developed a canonical RL method as well as two language model-based approaches. The first approach BaseLang mirrors AutoGPT [72], while the second, named Mastaba, is a hierarchical amalgamation of individual BaseLang models. We present the preliminary results of our initial endeavors. The RL agent demonstrates reasonable performance in mini-games. However, for both RL- and LLM-based agents, substantial progress in the full game remains a challenge.

In summary, this paper makes three major contributions. First, we introduce an interactive environment for a Civilization-based game that poses new challenges in both learning and reasoning, enriched with mini-games and support for two distinct types of APIs. Second, we outline the design of baseline methods and present a language model-based approach. Third, we share the preliminary results of our explorations, laying the groundwork for further research in this landscape.

## 2 RELATED WORK

**Complex Interactive Environments**. Interactive decision-making environments have exhibited increasing complexity with two key dimensions. Task complexity has grown in both physics-based and game environments, transitioning from simpler forms like MuJuCo [76] and ALE [6] to more realistic [1; 54; 59; 66; 44] or more intricate open-ended settings (*e.g.*, MineCraft [37; 18] and so on [2; 39; 38; 64]) that feature partially observable worlds and tasks with real-world resemblances. However, unlike *CivRealm*, their state and action spaces typically remain static during gameplay.

Simultaneously, the complexity has expanded from single-agent scenarios to multi-agent challenges. Multi-agent environments introduce additional difficulties due to non-unique learning objectives,

Table 1: Comparison with existing environments. *CivRealm* features the following characteristics for learning and reasoning. **Imperfect info:** The full state is partially observable. **Stochastic:** The dynamics of the environment is non-deterministic. **Multi-goal:** There are multiple victory conditions in the game. **Dynamic space:** The state and action space of a single player change dynamically in a game. **Multi-agent:** There are multiple interacting players in the game. **General-sum:** It is a mixed motive game, where cooperation and competition coexist. **Changing players:** The number of players can increase or decrease during a single game. **Comm.:** players can explicitly communicate in the game. **Tensor & Lang.:** The environment provides both tensor and language APIs.

| Environment | Imperfect info | Stochastic | Multi-goal | Dynamic space | Multi-agent | General-sum | Changing players | Comm. | Tensor & Lang. |
|---|---|---|---|---|---|---|---|---|---|
| MineDojo [18] | ✓ | ✓ | ✓ | ✓ | ✗ | ✓ | ✗ | ✗ | ✓ |
| MPE [50] | ✗ | ✓ | ✗ | ✗ | ✓ | ✓ | ✗ | ✓ | ✗ |
| Hanabi [5] | ✓ | ✗ | ✓ | ✗ | ✓ | ✓ | ✗ | ✓ | ✗ |
| Hold'em [10] | ✓ | ✗ | ✗ | ✗ | ✓ | ✗ | ✗ | ✗ | ✗ |
| Diplomacy [51] | ✗ | ✗ | ✗ | ✗ | ✓ | ✗ | ✓ | ✓ | ✓ |
| Melting pot [43] | ✓ | ✓ | ✓ | ✗ | ✓ | ✓ | ✓ | ✗ | ✗ |
| Google Football [42] | ✗ | ✓ | ✗ | ✗ | ✓ | ✗ | ✗ | ✗ | ✗ |
| Stratego [53] | ✓ | ✗ | ✗ | ✗ | ✓ | ✗ | ✗ | ✗ | ✗ |
| SMAC [58] | ✓ | ✗ | ✗ | ✗ | ✓ | ✗ | ✗ | ✗ | ✗ |
| Dota 2 [8] | ✓ | ✓ | ✗ | ✓ | ✓ | ✗ | ✗ | ✗ | ✗ |
| StarCraft II [79] | ✓ | ✗ | ✗ | ✓ | ✓ | ✗ | ✗ | ✗ | ✗ |
| *CivRealm* | ✓ | ✓ | ✓ | ✓ | ✓ | ✓ | ✓ | ✓ | ✓ |

non-stationarity, and diverse information structures [90; 84]. Examples such as MPE [50], Melting Pot [43], Neural MMO [70], Stratego [53], Diplomacy [51; 17], and StarCraft [79; 80], each offer distinct perspectives on the complexities of multi-agent decision-making. In comparison, *CivRealm* stands out for its comprehensive features for generalization, characterized by ever-changing in-game conditions. For a more detailed comparison, please refer to Table 1. In summary, *CivRealm* offers a platform for assessing decision-making agents' reasoning abilities at a broader scale.

**Reinforcement Learning Agents with Reasoning**. Reinforcement learning (RL) has witnessed significant growth with a focus on model-free methods [47; 61; 62; 22; 80]. As a culmination, AlphaStar [80] achieved mastery in StarCraft II. However, it still has generalization issues, relying heavily on human demonstrations and prolonged training. To overcome these challenges, knowledge representation and reasoning are crucial as shown in early work on *Civilization* [12; 11]. To integrate RL and reasoning, typical ways include search-based methods that select actions based on potential outcomes [13; 68; 69; 60], model-based RL that learns a dynamic model of the environment [49; 36; 86; 40; 23; 24; 26; 67; 28], and hierarchical RL that employ temporal and state abstractions [35; 78; 63; 16; 25; 3]. Despite these advances, solving highly complex tasks remains difficult for RL agents. *CivRealm* **comprehensively benchmarks these agents' generalization ability.**

**Large-Language-model-based Agents with Learning**. In the past two years, there has been significant progress in the development of LLM-based agents for task planning and reasoning in (multi-agent) interactive environments [14; 55; 87; 46; 89; 52; 56; 31; 34; 33; 29; 45; 27]. These agents have evolved from not incorporating learning [82; 93; 91] to becoming more adaptive with memory [71], self-reflection mechanisms [85; 65; 48], and learning capabilities [81; 92; 88; 65]. Nonetheless, research has revealed certain shortcomings in the reasoning capabilities of LLMs [7; 77; 9; 32]. **Given the extensive human knowledge LLMs have acquired,** *CivRealm* **is an ideal environment to assess their ability to learn from interactions and apply knowledge for reasoning.**

## 3 ENVIRONMENT

In this section, we begin by introducing the open-ended characteristics of *CivRealm*, followed by its engineering features. Subsequently, we describe both the full game and the mini-games supported by *CivRealm*, along with a discussion of their research values for building decision-making agents.

There are various characteristics of this environment that make it open-ended (Table 1): **Imperfect information**, where players only have access to information discovered by their own units and cities. **Stochastic dynamics** with random events and crises that can disrupt plans. **Multiple victory paths** are possible (*e.g.*, conquer, science, or highest score), requiring a balance between economic expansion, military development, diplomacy, culture, and technology. **A dynamic game space** with continuous changes in state and action space for a single agent. **Multi-agent interactions** with built-

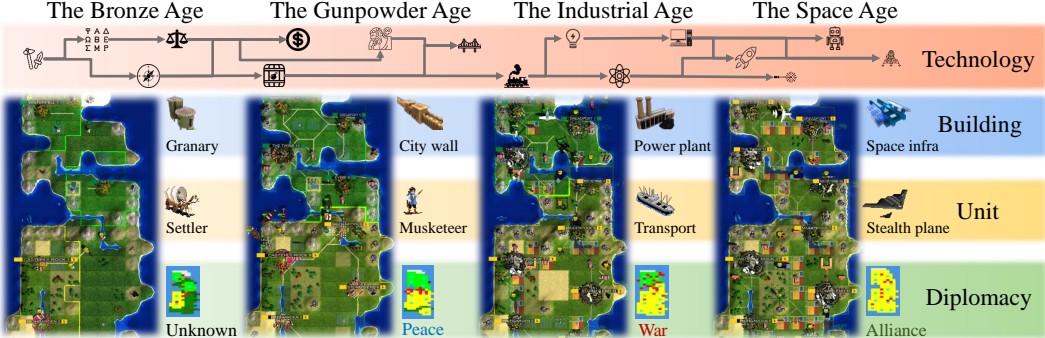

Figure 2: Civilization evolves as the game unfolds, and the potential state and action space explode. This figure focuses on 4 of the 8 ages, wherein technological advancements unlock a greater number of buildings and units. Throughout the course of the game, the state can grow from $10^{15}$ to $10^{650}$, and the action space can expand from $10^4$ to $10^{166}$ (§ D). This figure only shows some example elements; the full game includes 87 types of technologies, 68 types of buildings, 52 types of units, 6 government types, and 5 diplomatic states, all subject to the rule sets used and are customizable.

in AI players or other models, providing the potential for self-play. **General-sum** game that allows alliance formation during gameplay, which changes the game structure and makes the victories of different players non-exclusive. **Changes in the number of players** during a game due to revolts or conquers, leading to significant alterations in the joint state and action space. **Communication** between players through diplomatic actions and natural language chat, allowing agents to use their natural language capabilities. In summary, *CivRealm* presents unique challenges and complexities, making it an open-ended testbed for decision-making agents. Please see § A.1 for more details.

**Agent-architecture-agnostic framework**. *CivRealm* empowers each agent to act as a player in the open-source turn-based strategy game Freeciv [74]. *CivRealm* employs a server-proxy-client framework and implements proxy APIs so that a server hosts the game and the proxy establishes the connection between agents (*i.e.*, clients) and the server. The proxy distributes the game states received from the server to each agent and submits the actions returned by agents to the server. By this design, agents with various architectures can seamlessly engage in Freeciv by interpreting the observations provided by the proxy and generating actions that adhere to *CivRealm*'s specifications. **LLM-based-agent-friendly**. Freeciv is a turn-based game that operates without the need for real-time reactions. This affords players ample time for thoughtful deliberation. This pace aligns well with the operation of LLM agents, which typically demand substantial time for inference. **Evaluation platform for generalization ability**. *CivRealm* offers multiple convenient methods to create novel scenarios, such as generating random maps with diverse landscapes and varying player and unit numbers, or modifying the rule sets that define the fundamental game rules. These elements result in new configurations, demanding agents to reason the underlying game mechanics rather than relying solely on memorized experiences and public knowledge. Therefore, *CivRealm* serves as an effective platform for assessing the generalization capabilities of decision-making agents. **Support for a variety of tasks**. *CivRealm* offers a wide range of learning and reasoning tasks. These tasks include not only the comprehensive full game of Freeciv, but also smaller-scale mini-games designed using *Lua* scripts. In § 3.2, we will provide detailed descriptions of these tasks.

## 3.1 FULL GAME DESCRIPTION

In *CivRealm*, players take the role of civilization leaders with the objective of guiding their civilization from its humble beginnings to greatness, where full games can last from several hours to several days. Civilizations evolve through eras, with an explosion in the number of controllable objects as the game progresses, resulting in vast state spaces and joint actions (Figure 2). Decisions in the game have multifaceted impacts, encompassing both long-term strategic consequences and short-term tactical outcomes. This complexity necessitates a thought process that carefully weighs the implications of these decisions since greedy moves can easily be non-optimal in the long term.

**Observations**. Instead of directly processing raw pixel data of the game interface, we extract representative discrete information from graphics observed during human gameplay. These observations encompass data related to the map, units, cities, government, technology, and diplomacy. The *map*

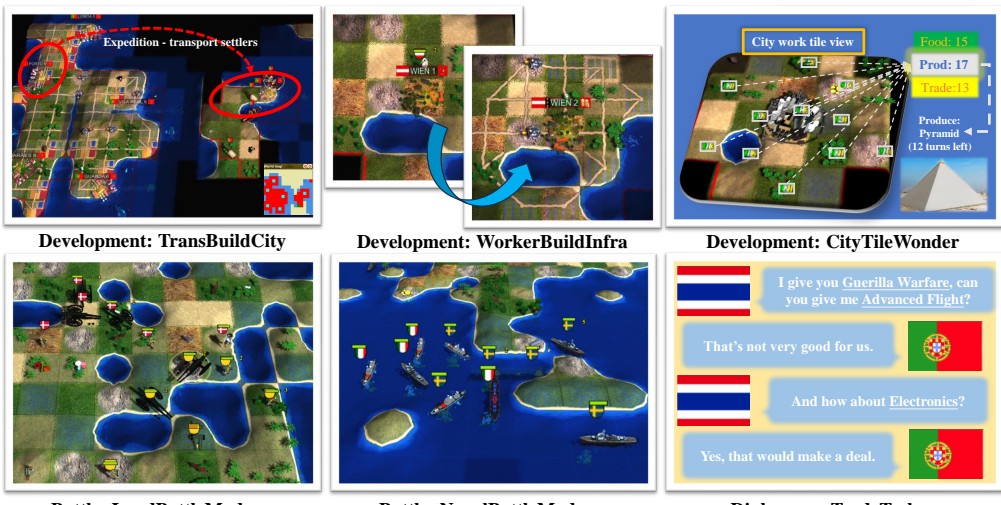

Figure 3: Examples of different types of the designed mini-games.

information includes whether a particular tile is observable, where a "tile" refers to a square space on the grid-based map. The map provides details like the terrain type, owner of the tile, resource output, additional infrastructure, and units present on the tile. The *unit* information provides insights into a unit's health, location, owner, attack/defense strength, remaining movement points (indicating the actions the unit can take in a turn), maintenance costs, *etc*. The *city* information covers details such as a city's location, owner, size, population, shield value, resource production, and more. The *government* information indicates the current government type of the civilization, the tax rate, *etc*. The *technology* information displays the technologies that have been researched and the technology currently being researched. The *diplomacy* information comprises data regarding diplomatic relationships with other players. For a comprehensive list of these observations, please refer to § A.1.1.

**Actions**. We have implemented a rich set of action classes that encompass the five primary facets of gameplay: *unit*, *city*, *government*, *technology*, and *diplomacy*. The *unit* actions are responsible for controlling a player's units. They can be categorized into three main types: engineering actions, which handle tasks like city construction, planting, mining, and more; movement actions, including moving, transportation, embarking, and so on; and military actions, such as attacking, fortifying, bribing, *etc*. The *city* actions pertain to the development and management of a city. They include unit production, building construction, city worker assignment, and more. The *government* actions allow players to change their government type to gain corresponding political benefits, adjust tax rates to balance economic expansion and citizen happiness, *etc*. The *technology* actions enable players to set immediate or long-term goals for their technology research. The *diplomacy* actions empower players to initiate negotiations, such as trading technologies, negotiating ceasefires, forming alliances, *etc*. For an exhaustive list of the implemented actions, please refer to § A.1.2.

**Evaluation Metrics**. *CivRealm* offers evaluation metrics to assess playing performance across various dimensions, including population, constructed cities, researched technologies, produced units, explored land, *etc*. An aggregated score is provided for overall evaluation. Please refer to § A.1.3.

## 3.2 MINI-GAME BENCHMARKS

*CivRealm* requires players to balance multiple factors, including economic expansion, military development, diplomatic influence, cultural achievements, and technological research, when making decisions. To provide a comprehensive benchmark and support for various learning paradigms (*e.g.*, curriculum learning), we built a collection of mini-games that focus on specific aspects of the full game. These mini-games are generated using the same rule sets as the full game, but with a smaller map size and fewer players. The mini-games are designed to be challenging, requiring agents to master specific skills to achieve victory. The key features of the benchmark are as follows.

**Unlimited task generation**. The benchmark offers an automatic mini-game generation tool, which introduces randomness across various aspects, including the map landscape, initial position of units and cities, tile resources, *etc*. Consequently, this provides access to an inexhaustible pool of mini-

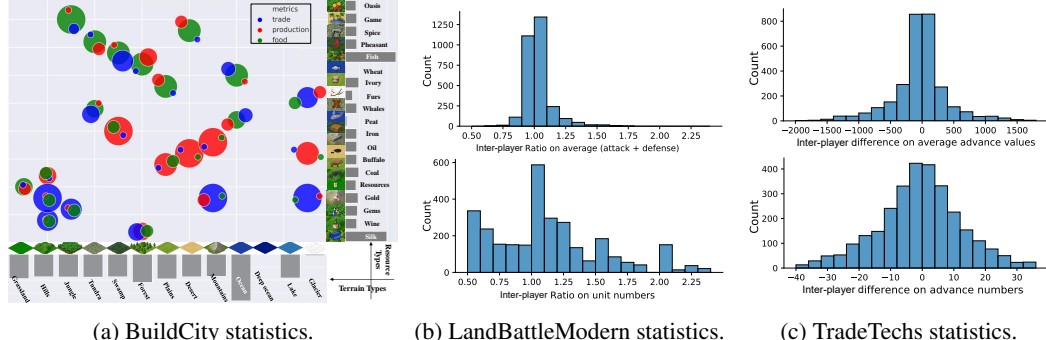

| (a) BuildCity statistics. | (b) LandBattleModern statistics. | (c) TradeTechs statistics. |

Figure 4: The generated mini-games are diverse and balanced. (a) Terrain and resource distribution and corresponding joint food/production/trade metrics. (b) Unit numbers and strength: inter-player ratio's distribution. (c) Technology number and value: inter-player difference's distribution.

games, allowing the assessment of agents' generalization abilities in smaller-scale tasks.

**High customizability**. The benchmark utilizes *Lua* scripts to specify the reward structure of mini-games conveniently. This modular design enables users to customize the victory condition of mini-games and train algorithms with different paradigms (*e.g.*, curriculum learning and meta learning).

**Compatibility with full game**. Mini-games adhere to the same input and output format as the full game. Agents trained in mini-games can be seamlessly migrated to the full game, and vice versa.

In *CivRealm*, we design **10 types of mini-games** and use the generation tool to produce 10,000 instances for each type. They can be categorized into three groups: development, battle, and diplomacy (as exemplified in Figure 3). The *development* mini-games aim at nurturing cities, focusing on factors like population growth, production efficiency, and economic prosperity. These elements are closely linked to a civilization's overall strength and advancement. The *battle* mini-games revolve around tactical warfare between opposing groups of units. Skilled commanders aim to defeat their enemies while minimizing their own losses. The *diplomacy* mini-games represent a unique feature of civilization games. Mastering diplomatic skills is pivotal, as maintaining positive relationships with other players can significantly influence the course of gameplay. Please refer to § A.2 for specific game definitions and the criteria we use to determine victory in these games.

**Task variation statistics.** The benchmark manipulates crucial configurations in each mini-game to increase the diversity of mini-game instances (§ A.2.2). In *development* mini-games, we introduce variations in terrain types and extra infrastructures on tiles, both of which impact resource outputs (food/production/trade). In *battle* mini-games, we alter the number of units and their attack/defense strengths. In *diplomacy* mini-games, we adjust the number and type of technologies held by each player. Figure 4 illustrates the benchmark effectively generates a diverse set of mini-games.

**Task difficulty level.** Owing to the varied configurations of mini-game instances, we classify their difficulty into three levels: *easy*, *normal*, and *hard* (§ A.2.1), based on whether the player controlled by our agent holds an advantage or disadvantage compared to the opposing player.

## 4 METHODS

We design three approaches as baseline methods, including: 1) a canonical tensor-based RL method inspired by AlphaStar [80], 2) an LLM method BaseLang that works similarly as AutoGPT, and 3) Mastaba, a hierarchical amalgamation of individual BaseLang models. In this section, we introduce these models as well as the challenges encountered. For more discussion, please see § E.

### 4.1 TENSOR-BASED REINFORCEMENT LEARNING

**Challenges** in *CivRealm* for tensor-based RL methods include the following. *Complex dynamics*: *CivRealm*'s complex game mechanics mimic human society, posing difficulties for tensor-based learning. It involves diplomacy, economics, technology, and military strategy, making it hard for both model-free and model-based approaches. *Overwhelming information*: agents face extensive information from units, cities, players, *etc*, while lacking semantic understanding. *Multi-Level actions*: *CivRealm*'s action space is complex and hierarchical, requiring sequential decision-making and neural network adaptation. *Sparse, delayed rewards*: Rewards are sparse, delayed, and asynchronous due to the turn-based and society-simulating nature *CivRealm*. *Diverse victory paths*:

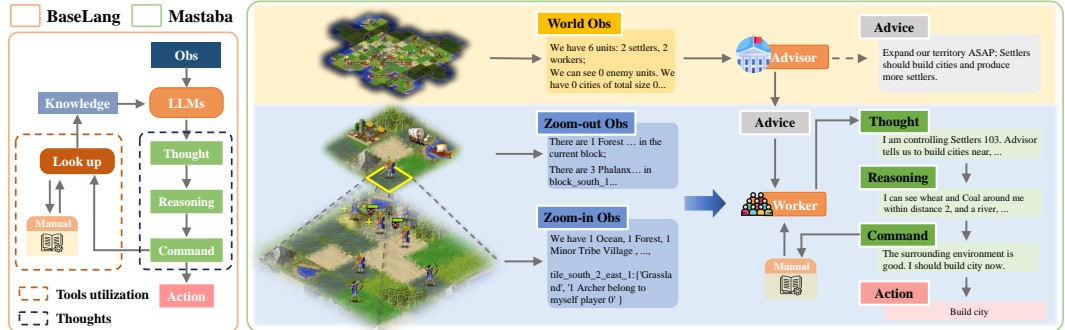

Figure 5: Architecture of LLM-based approaches. Left: BaseLang. Right: Mastaba.

*CivRealm* offers multiple paths to victory, complicating RL training that relies on reward signals.
**Network Design** We implement a hierarchical approach inspired by AlphaStar [80] to handle diverse inputs effectively. Our network has three main parts (Figure 11). *Representation*: we use MLP, Transformer, and CNN models to extract features based on input type (vector, sequence, or image-based). These features are globally connected through a transformer, and an RNN combines current-state features with a memory state for historical context. *Action Selection*: we leverage the learned representations to make decisions. The actor module selects the primary action category (e.g., unit, city, government, or turn termination), followed by a pointer network choosing the specific action ID. *Value Estimation*: we include a value prediction head to enable actor-critic learning, sharing the representation for training efficiency. We train the network using Proximal Policy Optimization [62] and parallelize tensor environments with Ray [4] (see § E.1.3 for training details).

## 4.2 BASELANG: BASELINE LANGUAGE-BASED AGENT

**Necessity and challenges** The need for developing an LLM-based agent in *CivRealm* arises from two factors. 1) LLMs are capable of task generation and problem-solving, augmented by their extensive human knowledge base. Their capabilities and knowledge are promising for solving decision-making problems. 2) LLM is not yet sufficiently sophisticated for complex reasoning [77; 7]. In *CivRealm*, LLM is advantageous due to its ability to use natural language, prioritize strategic gameplay, and handle diplomatic interactions. However, constructing such an agent faces several challenges, including managing multiple in-game roles, handling sparse and complex observations, addressing the long-term impact of actions, and improving the agent's decision-making over time.
**Design** We design the baseline language-based agent composed of three components (Figure 5): observation, reasoning, and commands. For observation, a 5x5 tile-based observation is employed, centered on each unit's location, optimizing tactical information provision while accommodating strategic depth. The reasoning module mimics AutoGPT [72] and outputs in three stages: thought, reasoning, and command. Commands empower the agent with the choice between "manual and history search" and "final decision" commands, enabling data retrieval from a vector database or selecting available actions to execute based on environmental and historical context. Finally, individual LLMs are assigned to each unit, with their context histories, to facilitate detailed planning.

## 4.3 MASTABA[1]: ENHANCING BASELANG BY A HIERARCHICAL STRUCTURE

To facilitate cooperation between independent entities, Mastaba introduces a hierarchical structure, organizing LLM workers, observations, and decision-making into a pyramid-like structure.
**LLM Workers**. Within Mastaba, LLM workers are structured as two layers. At the pinnacle is the "advisor", tasked with overseeing all other LLM workers. The advisor monitors the holistic nationwide perspective, including unit counts, city metrics, and enemy/player information. At the operational level, Mastaba maintains LLM workers that resemble BaseLang's structure.
**Observation**. Mastaba adopts a pyramid-like map view, condensing data from a $15 \times 15$ tile region into 9 blocks, each spanning $5 \times 5$ tiles. This design enables entities to grasp information within a broader range while managing prompt loads effectively, thereby elevating map awareness.
**Decision-making**. Mastaba's decision-making workflow follows its agent structure. The advisor

---

[1]Ancient Egyptian tomb before pyramids. The first pyramid in Egypt is considered a stack of Mastabas.

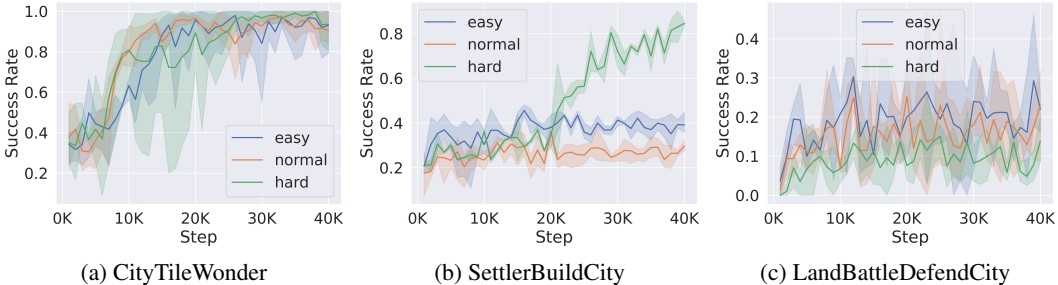

(a) CityTileWonder      (b) SettlerBuildCity      (c) LandBattleDefendCity

Figure 6: Performance of the RL method over training on three mini-tasks; the diverse mini-tasks exhibit varying degrees of stochasticity and complexity that lead to different success rates. In the "SettlerBuildCity" task, the RL method exhibited shortcut learning [20] in hard level.

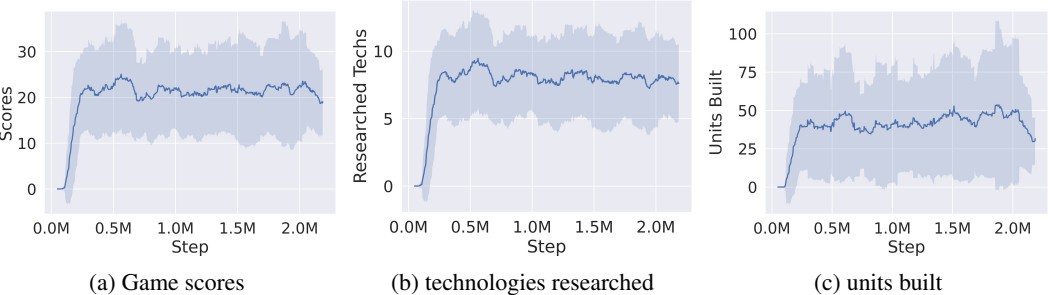

(a) Game scores      (b) technologies researched      (c) units built

Figure 7: Scores over the training course on the full game for the RL method. *CivRealm* provides different metrics to gauge the performance of the model from different aspects to provide further insights; these metrics also reflect the multi-goal nature of the environment (§ A.1.3).

initiates each turn with a nationwide assessment, encompassing cities, units, and potential threats. It generates suggestions during each turn and communicates them to other LLM workers, who independently select actions for their entities. Additionally, workers possess the capability to query a vector database for knowledge, enabling informed decisions based on manual or stored experiences.

## 5 EXPERIMENTS

### 5.1 TENSOR-BASED REINFORCEMENT LEARNING

To evaluate the performance of our tensor baseline across various mini-tasks and full games, we kept the same architecture and training hyper-parameters and trained our models from scratch.

**Minitask** For each minitask of a given difficulty level, we trained tensor baseline models for over 40,000 steps with 3 random seeds. Figure 6 shows 3 representative tasks with the success rates. While a small number of *development* minitasks, such as 'CityTileWonder', could be solved with a high success rate ($\sim 90\%$), most minitasks still pose a hierarchy of increasingly difficult challenges to our tensor baseline. These results highlight the strengths and limitations of current tensor-based RL models in handling different types of tasks in *CivRealm*. Specifically, RL models exhibit better learning capabilities for tasks offering immediate rewards and requiring relatively short-term planning. However, when confronted with tasks with sparse and delayed rewards, requiring long-term strategic planning, these models encounter significant difficulties in identifying viable solutions.

**Fullgame** For full games, we trained tensor baseline models for over 2 million steps. The overall performance is measured by the average score at the end of each game as shown in Figure 7. The score is a multi-facet metric calculated from a weighted sum of important game stats including population, technology researched, units built/killed, *etc*. Our RL model exhibits progressive learning, primarily focusing on unit production to enhance its score. However, it is important to note that this approach is myopic in nature. A more strategic and long-term approach involves prioritizing the establishment of additional cities to foster exponential growth in both technology and economics, although this strategy results in an initial decrease in population (and subsequently the score) in the short term. These observations underscore the existing limitations of RL methodologies. In this context, *CivRealm* emerges as an ideal platform for future research endeavors.

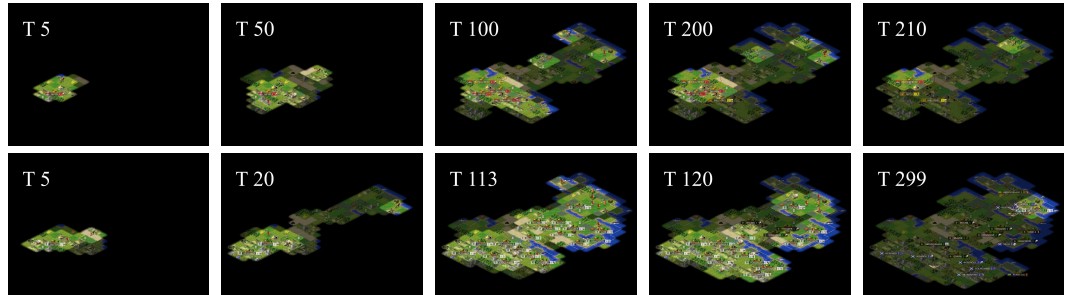

Figure 8: Evolution of the civilization. Top: BaseLang; Bottom: Mastaba. From left to right, this visual illustrates the progression of the empire through various phases: inception, establishment of a new colony, peak expansion of the realm, encountering invasion, and approaching collapse.

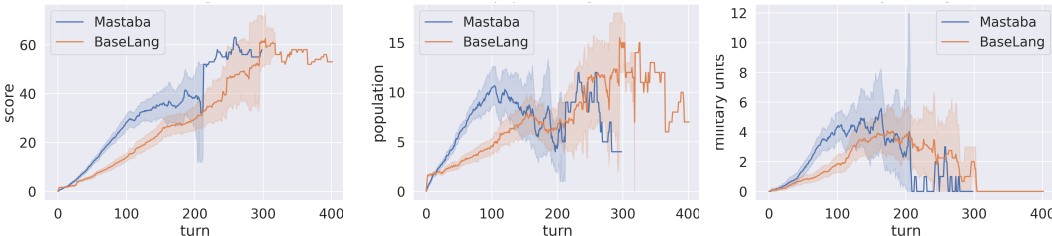

Figure 9: Stats on metrics of LLM-based agents against game turns for 50 individual games each.

## 5.2 LANGUAGE-BASED AGENTS: BASELANG AND MASTABA

We compared Mastaba with BaseLang on 10 maps with distinct geography. Each method plays 5 full games on each map. Figure 8 and Figure 9 show the qualitative and quantitative results, respectively. Note that one of the games lasts longer than the others, resulting in no variance in the later stage.

As depicted in Figure 9, Mastaba's development significantly outpaces that of BaseLang. Mastaba evolved with a larger population, a greater number of military units, and a higher score. Figure 8 shows Mastaba achieves a more extensive explored and settled area by turn 113, surpassing Base-Lang's progress by turn 210. In some games, BaseLang manages to survive longer than Mastaba. This is a consequence of Freeciv's built-in mechanics, which trigger pirate invasion when a player's score exceeds a certain threshold. Consequently, Mastaba encounters invasions at an earlier stage.

Our experiments offer valuable insights into LLM-based methods. Firstly, it is challenging to control a large number of objects (*e.g.*, units, cities) based on their local observations due to the context limit of LLM. BaseLang's sluggish expansion pace underscores the absence of a global perspective, whereas Mastaba's pyramid-like structure empowers the advisor to oversee the overall progress, establish a comprehensive context, and thus assist each local object in making well-informed decisions. The performance contrast between Mastaba and BaseLang highlights the necessity of a hierarchical decision architecture for tackling the complex scenarios presented by *CivRealm*.
Secondly, these results expose the weakness in LLM's grounding ability. Despite exposure to a vast amount of human knowledge, LLM-based agents still exhibit inefficiencies in their reasoning within *CivRealm*, which mirrors human society. Even with a global context, Mastaba manages to achieve only moderate economic expansion performance while failing to defend against pirate invasions. This signifies LLM's difficulty in discerning the most critical issues within the current context and its struggles to effectively balance various aspects of gameplay. Consequently, *CivRealm* serves as an ideal platform for evaluating and advancing the reasoning capabilities of LLM-based agents.

## 6 CONCLUSION

We present *CivRealm*, a distinctive challenge for decision-making agents, placing simultaneous emphasis on learning and reasoning abilities, which are evaluated across various mini-games and full games using diverse metrics. Experiments in *CivRealm* unveil limitations of contemporary methods, with tensor methods often yielding myopic strategies, and LLM-based agents still struggling with intricate reasoning tasks. To surmount these challenges, future research directions could involve integrating the strengths of RL and LLM, creating decision-making agents that combine adaptability and sophisticated reasoning. *CivRealm* serves as an ideal testing ground for these approaches.

ACKNOWLEDGMENT

The work of CivRealm is built upon several open-source projects, including Freeciv [74], Freeciv-web [75], FCIV-NET [73], and Freeciv-bot [57].

This work was supported by the National Natural Science Foundation of China. This work was also supported by Wuhan East Lake High-Tech Development Zone, National Comprehensive Experimental Base for Governance of Intelligent Society.

REPRODUCIBILITY STATEMENT

To help readers understand the environment details, we discuss the tensor-based API in § B, language-based API in § C, and method details in § E. For LLM experiments, we used GPT3.5-turbo provided by Azure's OpenAI API.

ETHICS STATEMENT

*CivRealm* can potentially contribute to the development of more socially aware AI agents, which could have applications in areas like policy-making, economic simulations, conflict resolution, and education. Moreover, the learning from such simulations could offer insights into the dynamics of human society, historical event outcomes, and potential future societal trajectories. Overall, the societal impacts of *CivRealm* and similar social simulation platforms can be profound, aiding in the development of AI that is more aligned with human values and societal goals.

The development of *CivRealm* is committed to fostering cultural and social inclusivity. It is developed upon the open-source game Freeciv [74], with a conscious effort made to incorporate a wide array of civilizations. Should any instances of marginalization or misrepresentation arise, we encourage and facilitate contributions to the source code for rectification.

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

# Appendices

## Content

# A ENVIRONMENT

## A.1 MORE ON FULL GAME AND *CivRealm* FEATURES

In the full game, each player acts as a civilization leader. The agents can be customized to be controlled by our agents or by the built-in AI algorithms. The objective of players is to guide their civilization from its humble beginnings to the greatest civilization and one full game can last from several hours to several days[2]. As depicted in Figure 2, civilizations evolve through eras from the Bronze Age to the Space Age and the number of controllable objects (units, cities, diplomatic relations, *etc.*) explodes as the game progresses. In addition, each decision made typically carries a multi-faceted impact, encompassing both long-term strategic consequences and short-term tactical outcomes. It is worth noting that a favorable tactical outcome may not necessarily translate into a positive strategic outcome. For instance, the immediate construction of a city at the beginning of the game can yield greater resources in the early stages (a tactical advantage). In contrast, settling in a resource-rich area after thorough exploration may result in substantial resource accumulation over the long haul (a strategic advantage). This complexity underscores the necessity for a learning and reasoning process that judiciously weighs the implications of long-term and short-term decisions.

Besides the long decision-making horizon, multi-faceted decision impacts, and huge state-action spaces, our environment exhibits additional characteristics (Table 1) that elevate its complexity:

**Imperfect info**. Players typically only gain the information discovered by their own units and cities, resulting in partially observable states. Players may also obtain others' vision by diplomatic actions.

**Stochastic**. The dynamics of the environment is stochastic. Moreover, there exist random events and crises that can disrupt plans, forcing players to adapt on the fly and make tough decisions.

**Multi-goal**. There are multiple victory paths, *i.e.*, (1) military: conquering all other civilizations; (2) science: being the first civilization that launches a spacecraft destined for Alpha Centauri; and (3) time: obtaining the highest score, computed based on criteria such as civilization size, wealth, cultural accomplishments, and scientific advancements, before reaching a predetermined number of turns, in case the first two conditions are not met. These paths necessitate a delicate balance between economic expansion, military development, diplomatic influence, cultural achievements, and technological research, which poses a high requirement for learning and reasoning.

**Dynamic space**. As the game unfolds, players continuously produce new units, construct additional cities, engage in battles, and conquer other players. Consequently, the state and action space of a player undergoes dynamic changes throughout the gameplay. Designing an effective decision model for the agent to adapt to this evolving space presents a significant challenge.

**Multi-agent**. Multiple players can interact with one another, including hand-crafted AI players provided by Freeciv on the server side. *CivRealm* allows multiple agents to connect to the same game simultaneously, facilitating self-play training.

**General-sum**. Players are free to form alliances or wage war against others, rendering the full game a general-sum game that necessitates considerations of both cooperative and competitive strategies.

**Changing players**. The number of players can fluctuate during a game due to factors like revolts or civilization conquests, introducing new players controlled by built-in AI or removing existing ones. Such changes often result in significant alterations to the state-action space.

**Comm.**. Players can communicate explicitly using two types of communication: diplomatic actions (*e.g.*, adding a clause) and natural language chat through a chat box. This feature enriches player interactions and enables LLM agents to fully leverage their natural language processing capabilities.

### A.1.1 OBSERVATIONS

In Freeciv, players primarily interact with a map composed of tiles where units and cities are situated. Additionally, Freeciv offers three tabs for conducting operations related to technology, diplomacy, and government. Consequently, the observations we extract from the Freeciv graphics encompass information pertaining to map tiles, units, cities, technology, diplomacy, and government.

---

[2]According to a poll: https://gamefaqs.gamespot.com/boards/938528-sid-meiers-civilization-v/66782899

*CivRealm* provides a set of feature layers for the observations of maps. We list these layers in Table 2, where $M \times N$ denotes the map size. Specifically, *status*, *type of terrain*, *owner of tiles*, *unit owner*, and *city owner* are layers with categorical values; *infrastructures* and *output* are layers with binary values; *unit distribution* layers are with scalar values. The upper bound of the unit number on one tile and the player number is 255, covering all possible scenarios in gameplay. The *status* being 0 means a tile has never been explored, being 1 means a tile has been explored before while being covered by war fog currently, and being 2 means a tile is within the vision range of the objects controlled by our agent. The *type of terrain* and *owner of tiles* denote the terrain type and owner of each tile, respectively. The *infrastructure* and *output* represent whether each tile has the corresponding infrastructure and resource output, respectively. The *unit distribution* means how many units of each specific type are on each tile. The *unit owner* and *city owner* record the owner of the units and city on each tile respectively.

Observations of a unit are listed in Table 3. For each unit in the game, if it belongs to other players, its features of *common unit field* can be known, while features of both *common unit field* and *my unit field* can be known if it belongs to our agent. For features of *my unit field*, its default value is set as -1 for units of others. The upper bounds of some features, which can be theoretically infinite, are set to very large values to cover all possible scenarios in gameplay. These settings also apply to the subsequent features related to diplomacy (Table 4), government (Table 5), city (Table 6), and technology (Table 7). Please refer to these tables for details.

### A.1.2 ACTIONS

We list the details of the actions supported by *CivRealm* in Table 11. These actions fall into five groups: unit, city, diplomacy, government, and technology. The *unit* actions demand precise control from players, enabling them to explore the map, execute tactical maneuvers, and construct infrastructure on the map. Additionally, there are miscellaneous unit actions we do not mention in the main text, including unit upgrading, trade route establishment, *etc*. The *city* actions empower players to produce units within their cities and enhance urban development to boost production outputs. The *diplomacy* actions facilitate the establishment of diplomatic relationships and the exchange of information, such as vision and technology. The *government* actions grant players the ability to instigate revolutions to change their government type and adjust tax, science, and luxury weights to balance various aspects of their civilizations. The *technology* actions enable players to define the direction of their technological development.

It is worth noting that certain actions are parameter-free, meaning their targets are self-evident (*e.g.*, increasing the tax rate). Conversely, other actions are parameterized, requiring additional parameters to specify their targets (*e.g.*, the embarking action necessitates specifying which boat the unit embarks on). Due to the dynamic change (*i.e.*, the number of units, cities, and players) in Freeciv, the parameter space of parameterized actions can also vary, which requires additional consideration when designing decision-making agents.

The specific design of the parameter space for unit actions also deserves a detailed explanation. It's essential to note that the game map can be of arbitrary size, and certain unit actions require specifying a target on the map. Consequently, if we were to allow unit actions to specify arbitrary targets anywhere on the map, the action space would become excessively vast. To address this challenge, we restrict units to specifying targets within a range of nine tiles. This range includes

Table 2: Observations of Map.

| Fields | Attributes | Value domains | Descriptions |
|---|---|---|---|
| Basic map | Status | $[0, 2]$ | |
| | Type of terrain | $[0, 13]$ | Size: $M \times N$ |
| | Owner of tiles | $[0, 255]$ | |
| | Infrastructures | 0 or 1 | 34 layers of size $M \times N$ |
| | Output | | 6 layers of size $M \times N$ for 6 output types |
| Units and city on each tile | Unit owner | | Size: $M \times N$ |
| | City owner | $[0, 255]$ | |
| | Unit distribution | | 52 layers of size $M \times N$ for 52 unit types |

Table 3: Observations of Unit.

| Fields | Attributes | Value domains | Descriptions |
|---|---|---|---|
| | X | $[0, M]$ | X-coordinate |
| | Y | $[0, N]$ | Y-coordinate |
| | Owner | $[0, 255]$ | Player the unit belongs to |
| | HP | $[0, 65535]$ | Health point of the unit |
| | Produce cost | | Cost needed to produce this type of unit |
| | Veteran | 0 or 1 | Whether the unit is veteran |
| | Can transport | | Whether the unit can transport other units |
| Common unit field | Unit type | $[0, 51]$ | One of 52 unit types |
| | Obsoleted by | | The unit type this unit can upgrade to |
| | Attack strength | $[0, 65535]$ | Affect the attack success rate |
| | Defense strength | | |
| | Firepower | | The damage of a successful attack |
| | Unit ID | | - |
| | Moves left | | Actions the unit can take in this turn |
| My unit field | Home city | $[0, 32767]$ | City supports this unit |
| | Upkeep shield | | |
| | Upkeep gold | | Resources needed to support this unit |
| | Upkeep food | | |

Table 4: Observations of Diplomacy.

| General | Attributes | Values | Descriptions |
|---|---|---|---|
| | Player ID | $[0, 255]$ | - |
| | Team | | |
| | Name | text | - |
| | Is alive | 0 or 1 | - |
| Common player field | Score | $[0, 65535]$ | - |
| | Turns alive | | How many turns the player has lived for |
| | Nation | $[0, 559]$ | - |
| | Embassy text | text | Describe if there are embassies between players |
| | Love | | Describe players' attitudes to others |
| My player field | Mood | 0 or 1 | Peaceful or Combat |
| | Diplomacy state | $[0, 6]$ | A categorical vector of my diplomacy states with other players: armistice, war, ceasefire, etc. |

Table 5: Observations of Government.

| General | Attributes | Values | Descriptions |
|---|---|---|---|
| Common government fields | Government ID | $[0, 6]$ | - |
| | Government name | text | |
| | Goal government | $[0, 6]$ | Goal of revolution |
| | Gold | $[0, 65535]$ | Gold in treasury |
| My government fields | Revolution finishes | | # turns for current revolution to complete |
| | Science | $[0, 100]$ | Government investment for each aspect. Sum to 100. |
| | Tax | | |
| | Luxury | | |

the eight neighboring tiles (north, northwest, west, southwest, south, southeast, east, and northeast), along with the tile on which the unit is currently located. For targets beyond this predefined range, units must first move to a neighboring position and then designate those targets. This approach ensures that we maintain a manageable unit action space.

### A.1.3 EVALUATION METRICS

*CivRealm* adopts game scores across 16 dimensions provided by the game engine to assess playing performance, including population, economics, production, cities, researched technologies, military units, wonders, research speed, land area, settled area, gold, units built, units killed, units lost, and

Table 6: Observations of City.

| General | Attributes | Value domains | Descriptions |
|---|---|---|---|
| | City name | text | - |
| Common city field | X | $[0, M]$ | X-Coordinate |
| | Y | $[0, N]$ | Y-Coordinate |
| | Owner | $[0, 255]$ | Player this city belongs to |
| | Size | | - |
| My city field | City ID | | |
| | Food stock | | |
| | Shield stock | | - |
| | Granary size | | |
| | Buy cost | | Cost to buy the undergoing production |
| | Turns to complete | | # turns to finish the current production |
| | Luxury | | |
| | Science | $[0, 32767]$ | |
| | Food | | |
| | Gold | | Resource outputs in each turn |
| | Shield | | |
| | Trade | | |
| | Bulbs | | |
| | City waste | | |
| | City corruption | | - |
| | City pollution | | |
| | Growth in | text | # turns for city population to grow |
| | State | $[0, 2]$ | City state: disorder, peace, etc. |
| | Production kind | $[0, 1]$ | Unit or building |
| | Production value | $[0, 67]$ | Unit or building type being produced |
| | People angry | | |
| | People unhappy | $[0, 127]$ | Number of people of each mood |
| | People content | | |
| | People happy | | |
| | Surplus food | | |
| | Surplus gold | $[-32768, 32767]$ | - |
| | Surplus shield | | |
| | Surplus trade | | |
| | Can build unit | | Binary vectors corresponding |
| | Can build building | 0 or 1 | to units or buildings |
| | Having Buildings | | |
| | Last completion turn | $[0, 32767]$ | Turn No. when city completed the last production |

Table 7: Observations of Technology.

| General | Attributes | Values | Descriptions |
|---|---|---|---|
| | Research name | text | - |
| Common technology fields | Researching | $[0, 87]$ | the technology being researched |
| | Tech of each type | 0 or 1 | If each technology has been researched |
| My technology fields | Bulbs researched | | Accumulated technology bulbs |
| | Tech upkeep | $[0, 65535]$ | Cost to keep current technologies |
| | Science cost | | - |
| | Researching cost | | |
| | Tech goal | $[0, 87]$ | - |
| | Techs researched | | Last researched technology |

units used, as well as an aggregated score for the overall performance evaluation. This aggregated score aggregates multiple factors, i.e., population, researched technologies, wonders, units built and killed, culture, and if spaceship criteria is achieved by the player.

Table 8: Definition of mini-games.

| Category | ID | Name | Introduction |
|---|---|---|---|
| Development | 1 | SettlerBuildCity | Move settler to suitable areas for building a city. |
| | 2 | WorkerBuildInfra | Command workers to build infrastructures for improving cities. |
| | 3 | CityTileWonder | Arrange work tiles to speed up producing a world wonder. |
| | 4 | TransBuildCity | Transport settlers by ships to another continent and build cities. |
| Battle | 5-9 | LandBattle[Ancient, Medieval, Industry, Modern] | Defeat enemy units on land tiles (units from various ages). |
| | 10 | LandBattleAttackCity | Conquer an enemy city. |
| | 11 | LandBattleDefendCity | LandBattleDefendCity against enemy invasion for a certain number of turns. |
| | 12 | NavalBattle | Defeat enemy fleet on the ocean (with Middle Times frigates). |
| | 13 | NavalBattleModern | Defeat enemy fleet on the ocean (with several classes of modern ships). |
| Diplomacy | 14 | TradeTechs | Trade technologies with another civilization. |

## A.2 MINI-GAME DETAILS

We provide the concrete definition of each mini-game in Table 8. The mini-games belong to three classes -*development*, *battle*, and *diplomacy*, covering a diversity of typical sub-tasks of the *Civilization* game. The *development* class of mini-games is closely related to city tile's combined resource outputs (food, production, and trade), while the *battle* class of mini-games focuses more on unit movement/attack operations. The player should be aware of the unit attack/defense/HP points to make full use of them. As for the last class of mini-game: *diplomacy*, it is about the art of trade and negotiation to achieve maximal interests for the player's civilization.

### A.2.1 VICTORY CONDITIONS, SCORE/REWARD SETTINGS AND DIFFICULTY LEVELS

Based on each mini-game's characteristics, we design a set of evaluation principles for them. For instance, we assess *development* mini-games by verifying whether the combined resource outputs (food, production, and trade) from city tiles surpass a predetermined threshold. The success of *battle* mini-games is determined by the successful annihilation of enemy units or the conquest/defense of designated cities. We determine the achievement of *diplomacy* mini-games based on whether the agent negotiates favorable agreements with other players, such as exchanging valuable technologies. For the details of reward settings and the computation of victory conditions, please refer to Table 9.

We additionally partition the mini-games into different difficulty levels (easy, normal, and hard) to better evaluate the models' capability as well as enable curriculum design in future work. Specific criteria for the difficulty level divisions can be found in Table 9.

### A.2.2 MINI-GAME RANDOMIZATION

To ensure sufficiently diverse and balanced mini-game instances, we conduct a set of randomization processing during mini-game generation. Table 10 gives exact perspectives on which randomization operations are performed for each type of mini-game.

## B TENSOR ENVIRONMENT

**Fix Space Size** Deep reinforcement learning algorithms generally demand a constant size of the observation space and the action space, but the numbers of units and cities vary through the gameplay of *CivRealm*, introducing inherent conflicts. To solve this problem, a tensor environment embeds size-varying observations into a large constant-size observation space by truncating excessive entities and masking out non-existent ones.

### B.1 OBSERVATION

The tensor environment provides API for tensor models. Observation of tensor environment is a big tensor including all observations of map, unit, city, diplomacy, government, and technology in § A.1.1.

Table 9: Score, victory, reward, and difficulty of mini-games.

| ID | Score Setting | Victory Condition | Stepwise Reward | Difficulty |
|---|---|---|---|---|
| 1 | $\rho = 0.4 \times \text{food} + 0.4 \times \text{product} + 0.2 \times \text{trade}$
$\rho_c = \sum_{top-6} \rho$ | $\geq q_{80\%}\left(\sum_{top-6} \rho\right)$ | $\delta \rho_{tile}$ | $hard, \quad \rho_c \leq 2.5$ |
| 2 | $\sum_{city} \rho$ | $\geq q_{80\%}\left(\sum_{city} \rho\right)$ | | $normal, \quad \rho_c > 2.5 \,\&\, \rho_c \leq 7$ |
| 3 | $\tau_{max} - \tau_B^W,$
$\tau_B^W$ is the number of turns to complete the wonder | $\tau_B^W == 0$ | | $easy, \quad \rho_c > 7$ |
| 4 | $N_{cb}^o$
$N_{cb}^o$ is the number of city built on another land | $N_{cb}^o > 0$ | | $hard, \quad \tau_{max} = 5$
$normal, \quad \tau_{max} = 10$
$easy, \quad \tau_{max} = 15$ |
| 5-9 | $\delta\left(N_{unit}\right) = N_{unit}^a - N_{unit}^b,$
$N_{unit}^a, N_{unit}^b$ are unit counts of player a and b | $N_{unit}^b == 0$ | $\delta\left(N_{unit}\right)$ | |
| 10 | $\delta\left(N_{city,unit}\right) = N_{unit}^a + N_{city}^a - N_{unit}^b - N_{city}^b$ | $N_{city}^b == 0$ | $\delta\left(N_{city,unit}\right)$ | $hard, \quad N_{unit}^b/N_{unit}^a > 1.1 \,\&\, N_{unit}^b/N_{unit}^a \leq 2$ |
| 11 | $\delta\left(N_{city,unit}\right) = N_{unit}^a + N_{city}^a - N_{unit}^b - N_{city}^b$ | $N_{city}^a > 0$ and $\tau == \tau_{max}$ | $\delta\left(N_{city,unit}\right)$ | $normal, \quad N_{unit}^b/N_{unit}^a > 0.9 \,\&\, N_{unit}^b/N_{unit}^a \leq 1.1$ |
| 12 | $\delta\left(N_{unit}\right) = N_{unit}^a - N_{unit}^b$ | $N_{unit}^b == 0$ | $\delta\left(N_{unit}\right)$ | $easy, \quad N_{unit}^b/N_{unit}^a > 0.5 \,\&\, N_{unit}^b/N_{unit}^a \leq 0.9$ |
| 13 | $\delta\left(N_{unit}\right) = N_{unit}^a - N_{unit}^b$ | $N_{unit}^b == 0$ | $\delta\left(N_{unit}\right)$ | |
| 14 | $N_{tech_get}$ | $N_{tech_get} > 0$ | | AI-skill level |

Table 10: Randomization setting of mini-games.

| ID | Terrain type | Terrain location | Resource type | Resource location | Unit type | Unit location | Unit number | City location | Tech degree |
|---|---|---|---|---|---|---|---|---|---|
| 1 | ✓ | ✓ | ✓ | ✓ | | ✓ | | | |
| 2 | ✓ | ✓ | ✓ | ✓ | | ✓ | ✓ | ✓ | |
| 3 | ✓ | ✓ | ✓ | ✓ | | ✓ | | ✓ | ✓ |
| 4 | | ✓ | | ✓ | | ✓ | | ✓ | |
| 5-9 | ✓ | ✓ | ✓ | ✓ | ✓ | ✓ | ✓ | | |
| 10 | ✓ | ✓ | ✓ | ✓ | ✓ | ✓ | ✓ | ✓ | |
| 11 | ✓ | ✓ | ✓ | ✓ | ✓ | ✓ | ✓ | ✓ | |
| 12 | ✓ | ✓ | ✓ | ✓ | | ✓ | ✓ | | |
| 13 | ✓ | ✓ | ✓ | ✓ | | ✓ | ✓ | | |
| 14 | | | | | | | | | ✓ |

## B.2 ACTION

There are five types of actors in the tensor environment, *i.e.*, unit, city, diplomacy, government, and technology. Each type of actor has its corresponding action space and the overall action space is the joint action space of all actor types. For an actor, each action in its action space is denoted by an action class name associated with parameters. For example, an action in action class *Go to a target tile* is denoted by the action key: *goto_num* where *goto* is the action class name and *num* is a parameter describing the direction, thus action *Go to the north tile* is denoted as *goto_1* where the parameter 1 corresponds to the north direction.

A chosen action returned to the tensor environment is a triplet including the actor type, actor ID, and the action key. For example, assuming action *Go to the north tile* is chosen for a unit whose ID is 121, then the action triplet returned to the tensor environment is *(unit, 121, goto_1)*.

## C LANGUAGE ENVIRONMENT

### C.1 OBSERVATION

Language environment provides API for language models. Observation of language environment is a dictionary containing a *world observation*, and the *observation* of each actor.

To understand the global situation and circumstances, world observation consists of multiple sentences summarizing the game board. It includes statistics on our units, cities, enemy units, enemy cities, and if we are in a time of peace or war, etc., for example, *"We have 10 units: 3 Warriors, 4 Workers, 1 Settlers, 1 Diplomacy and 1 Explorer. We can see 4 enemy units. We have 5 cities of a total size of 14. We can see 1 enemy city and 0 other cities. We are under attack."*.

Observation of each actor consists of the actor's name, a list of its available actions, a zoomed-out, and a zoomed-in observation dictionary corresponding to its macro- and micro-observations. The

Table 11: The actions supported by *CivRealm*

| Component | Action description | Parameter |
|---|---|---|
| Unit | Go to a target tile | the target tile |
| | Enter a hut in the target tile for random events | |
| | Embark on a target boat mooring in an ocean tile | the target boat unit |
| | Disembark from a target boat mooring in an ocean tile | |
| | Unload all units carried by the transporter unit | - |
| | Board a boat mooring in the city of the current tile | |
| | Deboard a boat mooring in the city of the current tile | |
| | Fortify in the current tile | - |
| | Attack the unit in a target tile | the target tile |
| | Bribe a unit of other players to join us | the target unit |
| | Conquer a city belongs to other players | the target city |
| | Sabotage a city belongs to other players | |
| | Steal technology from a city belongs to other players | |
| | Mine in the current tile | - |
| | Irrigate in the current tile | |
| | Build road in the current tile | |
| | Build railroad in the current tile | |
| | Plant trees in the current tile | |
| | Build a city in the current tile | |
| | Build airbase in the current tile | |
| | Build fortress in the current tile | |
| | Transform the terrain of the current tile | |
| | Pillage an infrastructure in the current tile | |
| | Cultivate the forest in the current tile into a plain | |
| | Upgrade the unit | - |
| | Disband the unit itself to save cost | |
| | Keep the current activity in this turn | |
| | Set the unit's home city as the city in the current tile | |
| | Join the city in the current tile (increase city population) | |
| | Sell goods in the target city's marketplace | the target city |
| | Investigate a target city belongs to other players | |
| | Establish embassy in a target city belongs to other players | |
| | Establish a trade route from the unit's home city to the target city | |
| City | Choose a working tile for city | the target tile |
| | Do not work on a tile | |
| | Buy building or unit | - |
| | Change the type of a specialist | type of the target specialist |
| | Sell a building | the target building |
| | Construct a building | |
| | Produce a unit | the target unit |
| Diplomacy | Start a negotiation | target player ID |
| | End a negotiation | |
| | Accept treaty | |
| | Cancel treaty | |
| | Cancel vision | |
| | Add a basic clause | target player ID + target basic clause type |
| | Add a trading tech clause | target player ID + giver ID + target technology ID |
| | Add a trading gold clause | target player ID + giver ID + how much gold |
| | Add a trading city clause | target player ID + giver ID + target city ID |
| | Remove a clause | target player ID + parameters of the target clause |
| Government | Revolution | the target Government |
| | Set rates of luxury + science + tax | Rates of luxury + science + tax |
| Technology research | Set a current research goal | the target technology |
| | Set a future research goal | |

zoomed-in observation dictionary corresponds to a mini-map, centered on the location of the actor, with a customized length and width. The zoomed-out observation dictionary is similar but corresponds to a larger sub-map, with a larger length and width, at the expense of granularity. Hence, the zoomed-in observation describes the detailed surroundings of the actor, while the zoomed-out observation represents its general perception of the distant surroundings. Instead of using coordinates that are not natural language friendly, we use tile_dir_num_dir_num or block_dir_num_dir_num

to describe a location related to the actor. For example, tile_north_1_east_1 means the tile whose coordinates are $(1, 1)$ related to the actor's location. The tile_south_1_west_2 means the tile whose coordinates are $(-2, -1)$ related to the actor's location.

Please refer to Figure 10 as an example of an actor's observation dictionary. Zoomed-in observation corresponds to a $5 \times 5$ mini-map centered on the actor's location, thus it contains 25 tiles. Zoomed-out observation corresponds to a $15 \times 15$ sub-map. In this example, we define that a block is made up of $5 \times 5$ tiles, thus the above $5 \times 5$ mini-map is the block where the actor is currently located, and the remaining tiles in zoomed-out observation make up the 8 surrounding blocks. Based on these settings, we get the observation of the actor. Each tile or block has a list containing its status, terrain, and the infrastructures on it. Besides, the information on units and cities, as well as their owners is also on the list. Unit and city owners are also associated with diplomacy tags, for example, units on *current_tile* belong to *myself player*; 3 cities in *block_north_1* belong to *an Alliance player*.

## C.2 ACTION

The action returned to the language environment is a triplet including the type of the actor, actor ID, and the name of the chosen action in *available_actions*. For example, in Figure 10, the type of the actor is *unit*, its actor ID is *121*, assuming that action *Go to the north tile* has been chosen, then the final action return to language environment is *(unit, 121, move North)*.

## D ESTIMATION OF STATE / ACTION SPACE SIZES

The following estimations focus on a temporal section of the game. However, viewing the game as a long-term planning task, the analysis may change a lot, but the result would also be of great complexity. If we consider that the building actions are temporal, our decision is not single-turn Markovian, the state space would be multiple times larger on the exponent.

**On Turn 5**. Under ruleset `Civ2Civ3`, each nation started with 2 Settlers, 2 Workers, and an Explorer. On turn 5, each Settler has acted at most 5 times, so it could appear at 121 possible places as itself, and could build a city at 81 positions. Each city originally had no less than 5 building options, and 24 working options. Each Worker, at the same time, could occur at similar positions, and when some work is done, the range it occurs will be smaller. Explorers could move 3 tiles per turn. So state space for 2 Settlers (or the cities they turned into) is about $\frac{1}{2}((121 + 81 * 5 * 24)^2 - 1521 * 5 * 24 - 121) \approx 5 \times 10^7$ large (square of single Settlers status, counted in cases: a city is not built / a city is built, with city options, then excluding all conflicting cases where cities are too close to each other. Halving means two Settlers are identical in practice). Workers has states about $\frac{1}{2}((121 + 81 * 3)^2 - 121 - 81 * 3) = 66066$. Explorer contributes $31^2 = 961$ states. So total state space is of size about $3 \times 10^{15}$.

With all units in their original form (no cities built), explorers can move in 9 directions and stay (9 moves in total), each Worker has 9 moves and two working options on each tile on average. Settlers have 9 moves and a `build city`. So in this case, the action space is of size 19800.

**With 100 units and 50 cities**. A simple observation is: on a map of $80 \times 50$ where half of the tiles are lands (2000), each unit has 1000 free places to stay (the other half are deep in enemies' territory). Thus units may contribute $1000^{100} = 10^{300}$ different states. For the 50 city locations, suppose each may find 10 options on average when they are built, then $10^{50}$ additional complexity becomes a factor. The cities' current improvement brings a lot more complexity. Given each city 20 possible improvements to build, the number of states for each city may go up to $2^{20} \approx 10^6$, providing $10^{300}$ states in total. Therefore, we may conclude that the state space is at least $10^{650}$ in size.

About the space of actions. 100 units bring at least $10^{100}$ actions from moving and sentry / defense / building. As for cities, with relatively high research levels, each city could produce at least 20 different units/improvements, thus the action space for cities is $20^{50} \approx 10^{66}$. In total, we would estimate the action space no smaller than $10^{166}$, even without considering the actions on `tech`, `research`, `tax rate`, `diplomacy`, *etc.*

# E  METHOD DETAILS

## E.1  TENSOR-BASED REINFORCEMENT LEARNING

### E.1.1  CHALLENGES

*CivRealm* presents several challenges for contemporary reinforcement learning methods, which can be summarized as follows:

**Complicated environment dynamics**. *CivRealm* has an intricate and multifaceted game mechanism that closely parallels the complexities of human society, making it an exceptionally challenging domain for both model-free and model-based tensor-based learning methods. The game's mechanics encompass a wide array of elements, including diplomacy, economics, technology, and military strategy, with decisions having far-reaching consequences akin to historical societal developments. This intricate interplay of factors mirrors the multifaceted dynamics of real-world human societies, rendering it challenging for tensor-based agents to effectively capture and navigate. Whether adopting model-free or model-based approaches, these agents struggle to interpret the nuanced interactions, complex dependencies, and evolving conditions within the game, underscoring the nature of this environment and the need for innovative strategies to tackle its dynamics.

**Overwhelming observation information**. One of the primary challenges faced by tensor-based RL agents is the assimilation of vast amounts of information in a manner that lacks semantic understanding of the tensor values. In the context of *CivRealm*, the observation space is exceptionally rich and expansive, providing intricate details about units, cities, players, governments, and more. However, not all of this information holds immediate relevance for effective decision-making during each turn. The dynamic, ever-changing nature of the information, coupled with its variable length, presents a formidable obstacle for the agent to discern the significance of individual components and to establish connections among different pieces of information. Moreover, the inherent dynamism of the game, involving fluctuating player numbers and evolving game conditions, further compounds the complexity of this challenge.

**Dynamic multi-Level action space**. *CivRealm*'s action space is complex and hierarchical, making it impractical to model actions uniformly. Instead, actions need to be decomposed into multiple levels, with decisions made sequentially for each component. This introduces design challenges for neural networks and the selection of optimal actions, as the agent must navigate through this multi-level action hierarchy.

**Sparse, delayed, and asynchronous rewards**. In the context of *CivRealm*, rewards are sparse, delayed (*e.g.*, it can take many turns to finish a building), and often asynchronous. The agent receives a reward only when specific events or tasks are accomplished, making it challenging to provide timely feedback for learning. Many crucial actions, such as exploration, do not yield immediate rewards, and even actions that can lead to rewards are not individually rewarded. Instead, rewards typically accumulate over a series of actions in multiple turns, making it difficult for the agent to attribute its actions to future rewards. Additionally, rewards are received only at the end of a turn, which may mislead the agent into favoring turn-ending actions over actions that contribute to long-term objectives.

**Diverse winning/termination conditions**. Achieving victory in *CivRealm* is multifaceted, with various development paths, including military, technological, and temporal strategies. The evaluation of the agent's state varies depending on the chosen strategy, which complicates RL training. Traditional RL methods heavily reliant on reward signals may struggle to adapt to these diverse pathways to success, where cooperation and competition coexist.

In summary, *CivRealm* poses significant obstacles for RL agents, necessitating innovative approaches to tackle the issues of sparse rewards, overwhelming information, complex actions, and diverse victory conditions. Addressing these challenges will be essential for creating intelligent agents capable of mastering the game of *CivRealm*.

### E.1.2  FORMULATION OF GAME AS MDP

We consider a discrete-time infinite-horizon Markov decision process (MDP) defined by a tuple $(\mathcal{S}, \mathcal{A}, p_0, P, \mathcal{R}, \gamma)$, where $\mathcal{S} \subseteq \mathbb{R}^{d_s}$ is the space of states, $\mathcal{A} \subseteq \mathbb{R}^{d_a}$ the space of actions, $p_0(s_0)$

the distribution over initial states $s_0$, $P(s_{t+1}|s_t, a_t)$ the transition function, $\mathcal{R}$ the reward function, and $\gamma \in (0, 1]$ a discount factor. $\pi(a_t|s_t; \theta)$ is the policy parameterized by $\theta \in \mathbb{R}^{d_\theta}$. The objective of an RL algorithm is to train a policy that maximizes the expected sum of discounted rewards: $J(\theta) = \mathbb{E}_{\tau \sim p(\tau;\theta)} \left[ \sum_{t=0}^{H} \gamma^t r(s_t, a_t) \right]$. It is worth noting that, in the context of turn-based game *CivRealm*, we consider timestep in the action level instead of the turn level. That is, every action taken within a turn will lead to a change in timestep, and turn-done is also considered to be an action.

### E.1.3 Network Design

To effectively handle multi-source and variable-length inputs, we draw inspiration from AlphaStar [80] and implement a serialized hierarchical feature extraction and action selection approach. This method involves generating layered actions and predicting value function outputs, and our neural network architecture comprises three main components: representation learning, action selection, and value estimation.

**Representation**. At the representation level, we adopt a hierarchical structure. In the lower layer, we extract controller features using various models like MLP, Transformer, and CNN, depending on whether the input is a single vector, sequence, or image-based. These extracted features are then fed into a transformer to facilitate attention across different entities, creating globally meaningful representations. Additionally, we utilize an RNN to combine the current-state features with the memory state, enabling conditional policy decisions based on the state history.

**Action selection**. At the action selection level, we leverage the learned representations to make decisions. In the actor selection module, we determine the primary action category to be executed, including options like unit, city, government, or termination. Subsequently, we employ a pointer network to select the specific action ID to be executed, followed by the determination of the exact action to be performed.

**Value estimation**. To enable the use of an actor-critic algorithm, we incorporate a value prediction head after the representation learning phase. This shared representation part of the network benefits both the actor and critic, enhancing training efficiency.

**Training**. We use the Proximal Policy Optimization (PPO) [62] algorithm to train the agent. To mitigate the on-policy sample complexity of PPO, we harness Ray [4] for parallelizing tensor environments, optimizing training speed and efficiency. We configured the actor update for 5 epochs, employing a clipped value loss with a clip parameter of 0.2, and using one mini-batch per epoch. The coefficients assigned to the entropy term and value loss were 0.01 and 0.001, respectively. The length of each episode was set at 125 steps, and we collected training data across 8 parallel environments. The learning rate for the Adam[41] optimizer was established at 0.0005, with an optimizer epsilon of 0.00001. These parameter settings were carefully chosen to maintain a balance between the effectiveness of learning and the stability of the algorithm.

### E.2 BaseLang: Baseline Language-based Agent

LLM's emergent capabilities grant it a vast foundation of human knowledge, empowering it in various areas such as task generation, open-world long-term planning, and solving complex problems.

In *CivRealm*, LLM has three distinct advantages. Firstly, our environment prioritizes long-term planning and strategic gameplay over low-level control, allowing the agent to interact with the environment in natural language rather than precise control actions. Secondly, Freeciv is a turn-based game, not requiring real-time interaction, providing ample time for LLM to engage in long-term planning. Thirdly, diplomatic operations are challenging for tensor-based algorithms as agreements are usually achieved through conversations, but natural language allows for direct diplomatic interactions. Additionally, LLM's knowledge encompasses a wide range of human civilization evolution knowledge, including history, warfare, politics, technology, *etc.*, making it more akin to human players in terms of prior knowledge.

### E.2.1 Challenges

Due to the high complexity of the Freeciv game itself, there are many challenges in the process of constructing a language-based agent, including:

**Multiple role-playing**. Different from agents for Minecraft, the agents for *CivRealm* must play different roles at a time: controller of different units with different locations and abilities, Mayer of different cities, leader of the nation deciding research directions and diplomatic strategies. A language agent must play all these different but related identities simultaneously in a turn.

**Sparse and complex observation**. In the early stages of the game, there are extensive unseen areas in the player's field of view. This makes it necessary for early-stage agents to first learn to efficiently explore the environment using powerful prior knowledge, posing a significant challenge in constructing these agents. Additionally, the vast map size (4000-tile maps for our full game setup) and complex information on each tile overwhelm the context window of all language models. Thus, LLM agents have to prioritize reading the map properly and efficiently.

**Long-term effect of actions**. Freeciv is a game that requires a strong focus on long-term strategy and planning, where early moves can have an impact on decisions hundreds of turns later. However, the context length limitation of LLMs restricts their analysis to short time-frames, and figuring out valuable action information within sparse reward temporal trajectory data is a significant challenge.

**How does the agent improve itself**. Training or fine-tuning LLM is a highly challenging task, and the lengthy time span of *CivRealm* significantly increases the difficulty of collecting high-quality data. Therefore, we can only choose to use an in-context approach, continuously enhancing LLM's decision-making ability based on experiential information gathered during interactions with the environment. However, extracting core information from such a complex game remains a challenge, making it difficult to let the agent improve itself effectively.

### E.2.2   STRUCTURE

We adopted an AutoGPT-like approach to build a baseline language-based agent. AutoGPT [72] is a memory-equipped automated LLM agent capable of making autonomous decisions in context. The following sections will delve into observation, reasoning, commands, and structure separately. ion.

**Observation**. Due to the various challenges mentioned above, we need to provide our language-based agent with a more effective natural language-based observation. While, at the strategic level, players require very detailed map information for effective long-term planning, for each specific unit, providing only the most basic information it needs can fulfill its tactical requirements. Specifically, we designate an area of 5x5 tiles, totaling 25 tiles, centered on the location of each unit, as the observation provided to the agent.

**Reasoning**. For the reasoning module, we largely follow the design of AutoGPT, employing three modules: thought, reasoning, and command, to analyze information. The thought module determines what actions the agent should take in the current state. The reasoning module analyzes the current information to understand what additional actions are needed to achieve the current thought and why. The command module summarizes the information from the previous two parts and presents specific steps for each operation, which in our case is a command to execute. The integration of these three parts forms the reasoning module of the language-based agent.

**Commands**. In our design, we allow language-based agents to choose between two commands in the planning stage: "manual and history search" and "final decision." For the former, we first store a document related to Freeciv in a vector database. The agent can then use this command to query the database in a way that retrieves relevant information from the document based on semantic similarity. As for the latter, the agent selects actions for the currently controlled unit based on the current environmental information and the historical context information.

We represent this observation information using JSON, where the keys represent the relative positions of each tile with respect to the current unit's tile. For example, "north_1_east_2" indicates a position one tile north and two tiles east from the current unit's tile. The value information for each tile corresponds to the resources present on that tile and distinguishes between friendly and enemy agents. We consolidate the information from these 25 tiles as the observation for the unit currently controlled by the agent. For unexplored areas, we will not provide tile information.

**Structure**. After constructing our language-based agent, we created such an agent for each unit, each having its own separate context history for planning and decision-making. In the initial design, we used one agent to operate all players, but due to the inherent context limitations of the language

model, it was not possible to perform very long-term planning for a unit's behavior. Therefore, we believed it was necessary to build a separate agent for each unit.

When the interaction information exceeds the maximum context limit, we use the Conversation-SummaryBufferMemory module of Langchain to summarize the historical information. In addition to this, we established a unit list within the system, taking turns allowing our language-based agent to provide the corresponding action decisions, thereby achieving planning and control for each unit.

### E.3 MASTABA: ENHANCING BASELANG BY A HIERARCHICAL STRUCTURE

BaseLang faced challenges due to the independent, isolated behavior of its entities. Mastaba builds upon BaseLang and succeeds in managing a nation in the game. Each entity operated independently with limited communication through observations, hindering efficient cooperation. To mitigate these issues, BaseLang constrained each entity's view to a $5 \times 5$ tile area, limiting long-range planning—a significant drawback when communication between entities is restricted. Mastaba attempts to resolve these problems by introducing a primary pyramid structure to organize observations, agents, and decision-making, where different layers of observation will be fed into different levels of LLM units for decision-making.

#### E.3.1 DESIGN OVERVIEW

**Hawk-Eye mapview**. Inspired by the concept of hawk-eye vision, we have designed a multi-layer observation. Recognizing that entities do not necessarily require detailed information beyond a local $5 \times 5$ tile area, we condense data from a $15 \times 15$ tile region into 9 blocks, each measuring $5 \times 5$ tiles. The central block represents the immediate local area, while the remaining eight blocks correspond to adjacent directions. This arrangement forms a $3 \times 3$ grid, as illustrated in Fig. 5. This pyramid-like structure strikes a balance between map complexity and the richness of observations.

**The role of the advisor**. In our agent organization, each entity is associated with an independent LLM instance for communication. Additionally, we introduce a crucial entity called the "advisor" to oversee all other instances. The advisor is responsible for monitoring fundamental game information concerning the entire nation. This includes data such as the total number of units (both military and non-military), the number of cities owned by the entity, and the quantities of units and cities belonging to both enemies and other players. Ultimately, the advisor receives a nation status report, indicating whether the entity is currently being invaded, conducting invasions, maintaining a state of relative peace, or experiencing a lack of communication. The advisor's primary role is to generate suggestions during each turn and disseminate them to all other agent prompts. This advisor entity plays a pivotal role at the apex of a hierarchical structure within the agent framework.

**Specialized worker units**. The structure of our worker units closely follows the BaseLang framework. In addition to the general worker prompt from BaseLang, we introduce three distinct types of specialists: Settlers, Workers, and Explorers. The key distinction lies in the examples provided within their instruction prompts, which are tailored to their respective unit types. For the general LLM worker, we continue to employ the instruction prompt from BaseLang for all other entities. Furthermore, we enhance the output accuracy by requiring each worker to reiterate the available actions in their "thoughts".

**Decision workflow**. Mastaba follows a decision-making workflow structured like a pyramid. During each turn, the advisor initiates the decision-making process with an overarching assessment of the nation, taking into account cities, units, and potential threats from enemies. This decision is then communicated to the prompts of all other worker units. Subsequently, each worker unit independently selects an action for the entity it controls. Workers also have the opportunity to query a vector database to obtain knowledge from manual or stored experiences. Following a query, a worker must make a decision regarding the action to be taken by the entity under its control.

## F MORE EXPERIMENT RESULTS

We provide more results of our tensor-based RL method for mini-games in this section.

```
info['llm_info']['unit'][121] = {
    'name': 'Warrior 121',
    'available_actions': ['keep activity', 'set HomeCity', 'upgrade', 'cancel order',
                          'move North', 'move East', 'move NorthWest'],
    'observations': {
        'minimap': {
            'current_tile': ['1 Desert', '1 Irrigation', '1 Road', '1 Railroad',
                             '2 Phalanx', '4 Warrior', '1 Chariot',
                             'unit owners are: myself player_0',
                             '1 cities of myself player_0'],
            'tile_north_1': ['1 Hills', '1 Mine', '1 Road'],
            'tile_south_1': ['1 Ocean'],
            'tile_east_1': ['1 Desert', '1 Irrigation', '1 Road'],
            'tile_west_1': ['1 Plains', '1 Irrigation', '1 Road'],
            'tile_north_1_east_1': ['1 Mountains'],
            'tile_north_1_west_1': ['1 Desert', '1 Irrigation', '1 Road'],
            'tile_south_1_east_1': ['1 Plains', '1 Irrigation', '1 Road', '1 River',
                                    '1 Workers', 'unit owners are: my self player_0'],
            'tile_south_1_west_1': ['1 Ocean'],
            'tile_north_2': ['1 Hills'],
            'tile_north_2_east_1': ['1 Forest', '1 Road'],
            'tile_north_2_west_1': ['1 Hills'],
            'tile_north_2_east_2': ['1 Forest', '1 Road'],
            'tile_north_2_west_2': ['1 Grassland', '1 Road'],
            'tile_south_2': ['1 Ocean'],
            'tile_south_2_east_1': ['1 Ocean'],
            'tile_south_2_west_1': ['1 Ocean'],
            'tile_south_2_east_2': ['1 Desert', '1 Road', '1 Railroad'],
            'tile_south_2_west_2': ['1 tile unexplored'],
            'tile_east_2': ['1 Hills'],
            'tile_north_1_east_2': ['1 Mountains'],
            'tile_south_1_east_2': ['1 Desert', '1 Irrigation', '1 Road', '1 River'],
            'tile_west_2': ['1 Ocean', '1 Whales'],
            'tile_north_1_west_2': ['1 Forest', '1 Road'],
            'tile_south_1_west_2': ['1 Ocean']
        },
        'upper_map': {
            'current_block': ['1 tile unexplored', '7 Ocean', '5 Desert', '3 Forest',
                              '1 Grassland', '4 Hills', '2 Mountains', '2 Plains',
                              '6 Irrigation', '1 Mine', '12 Road', '2 Railroad', '2 River',
                              '1 Wheat', '1 Whales', '1 Workers', '2 Phalanx', '4 Warrior',
                              '1 Chariot', 'unit owners are: myself player_0',
                              '1 cities of myself player_0'],
            'block_north_1': ['11 Forest', '4 Grassland', '7 Road',
                              '3 cities of a Alliance player_2'],
            'block_south_1': ['13 tiles unexplored', '7 Ocean', '1 Forest', '1 Grassland'],
            'block_east_1': ['10 Desert', '3 Forest', '1 Irrigation', '4 Mine'],
            'block_west_1': ['2 tiles unexplored', '7 Ocean', '1 Swamp', '1 Mine'],
            'block_north_1_east_1': ['15 Forest', '3 Workers'],
            'block_north_1_west_1': ['6 Forest', '4 Grassland'],
            'block_south_1_east_1': ['3 Ocean', '2 Desert', '8 Forest', '3 Hills',
                                     '2 Jungle', '1 Mountains', '2 Plains', '1 Mine'],
            'block_south_1_west_1': ['5 tiles unexplored', '12 Ocean', '5 Grassland']
        }
    }
}
```

Figure 10: A language environment observation example. The length and width of the mini-map and upper-map, as well as the size of the block, can be customarily set.

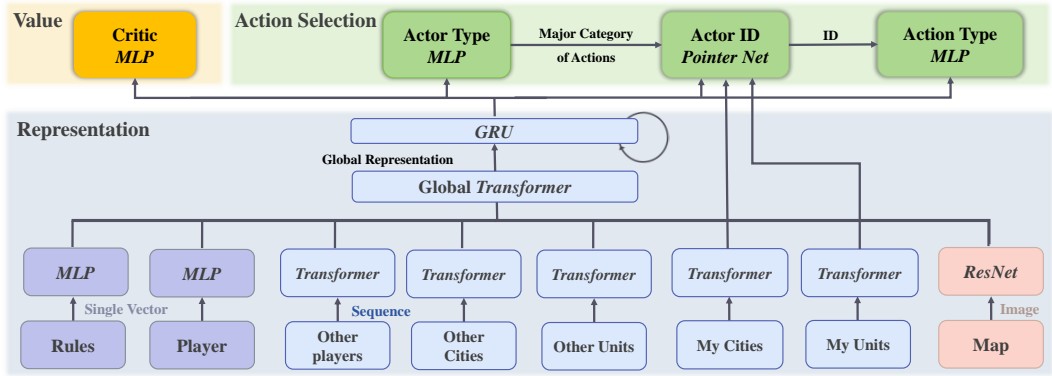

Figure 11: Network architecture of the tensor-based reinforcement learning agent.

Figure 12: Battle ancient era.

Figure 13: Battle attack city.

Figure 14: Battle defend city.

Figure 15: Battle industry era.

Figure 16: Battle info era.

Figure 17: Battle medieval.

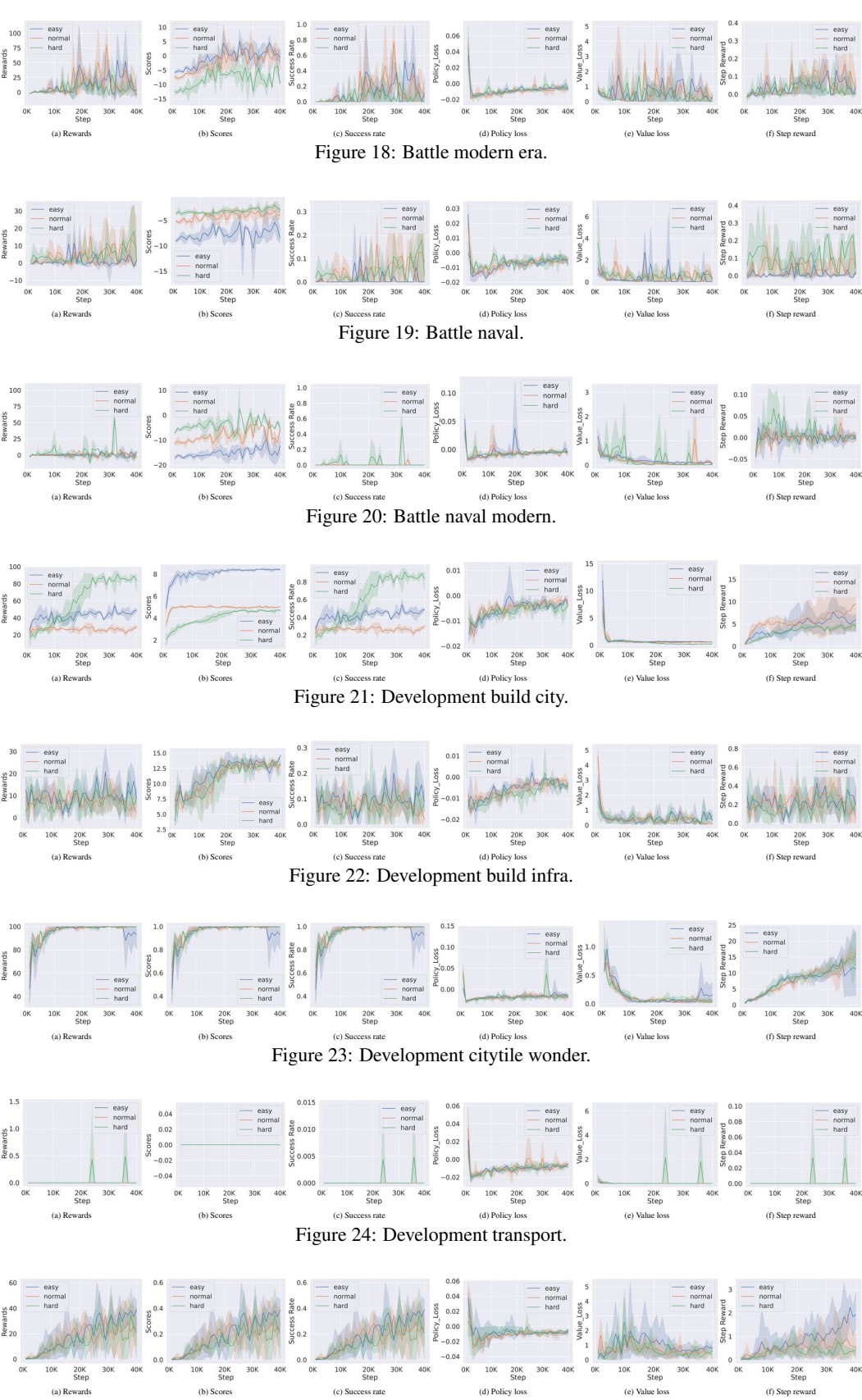

Figure 18: Battle modern era.

Figure 19: Battle naval.

Figure 20: Battle naval modern.

Figure 21: Development build city.

Figure 22: Development build infra.

Figure 23: Development citytile wonder.

Figure 24: Development transport.

Figure 25: Diplomacy trade tech.

