# OpenReview forum: "CivRealm: A Learning and Reasoning Odyssey in Civilization for Decision-Making Agents"
_ICLR.cc/2024/Conference — ICLR 2024 spotlight_

### Official Review · Reviewer_XJ4N · 2023-10-30

**Soundness:** 3 good
**Presentation:** 3 good
**Contribution:** 2 fair
**Rating:** 6
**Confidence:** 4

**Summary:**

The primary contribution of this paper is the introduction of the CivRealm environment, which is built on Civilization VI, a complex and popular video game. This environment offers a platform to study long-term planning, multi-agent interactions, and intricate decision-making etc.

**Strengths:**

- The paper doesn't just introduce an environment but also provides benchmark tasks and baseline.
- The API provides a comprehensive log of game state and actions.
- It allows AI agents to interact at both high (strategic) and low (tactical) levels.
- It focus on long-term planning which is crucial to agents

**Weaknesses:**

As a work in the area of datasets and benchmarks, I think a lot of the details are in the code, and I don't have good confirmation of those details just by virtue of the text in the paper. This environment is based on an open-source game http://freeciv.org/. I don't know how different is the open-source game and the environment introduced by the authors. It would be excellent if authors could give reviewers an early release of code to review.

**Questions:**

This environment is based on an open-source game http://freeciv.org/. From the development aspects, what is the main modifications/effort has been done upon the open-source game?

---

> ### Author Response · Authors · 2023-11-18
> **Point-to-Point Response to Reviewer XJ4N**
>
> We thank reviewer 8arZ for the valuable time and constructive feedback.
>
> #### **Q1: As a work in the area of datasets and benchmarks, I think a lot of the details are in the code, and I don't have good confirmation of those details just by virtue of the text in the paper. This environment is based on an open-source game <http://freeciv.org/>. I don't know how different is the open-source game and the environment introduced by the authors. It would be excellent if authors could give reviewers an early release of code to review.**
>
> **A1:**
> That is an excellent question. We have released our code on GitHub at an anonymous account for review purposes: <https://github.com/civrealm>. and you can also access our documentation at: <https://civrealm.github.io/civrealm/>. This GitHub account hosts four distinct repositories <https://github.com/civrealm?tab=repositories>:
>    - `civrealm`: this repository contains the core code of the CivRealm environment, encompassing both tensor-based and language-based APIs, along with comprehensive documentation.
>    - `civrealm-sav`: this repository contains the code responsible for creating the mini-games, which also enables users to customize their own games and generate new maps.
>    - `civrealm-tensor-baseline`: this repository provides the code for our baseline tensor-based RL agents.
>    - `civrealm-llm-baseline`: this repository hosts the code for our LLM agents, including BaseLang and Mastaba.
>
> In the following, we would like to explain the relationship between Civrealm and the Freeciv game. To put it briefly, it is analogous to the relationship between pysc2 ([link](https://github.com/google-deepmind/pysc2)) [ref1] and the Starcraft game.
>
> Specifically, Freeciv is a game that supports graphical interfaces for human players. However, this interface type is **not** designed for RL or language agents, i.e., accessing states and executing actions through tensor or language. To expose Freeciv as a research environment, we have developed CivRealm with an OpenAI Gym-style API to facilitate interaction between AI agents and Freeciv. In other words, Civrealm implements a proxy API that enables programmatic control of Freeciv. Through this API, AI agents can start/join a game, obtain game states, perform actions, and capture screenshots to generate game replays.
>
> To implement the API, we conducted an in-depth exploration of the internal workings of Freeciv. This involved:\
> (1) delving into the process of parsing game states and issuing action commands through communication packets, \
> (2) building concurrent communication flows and thoroughly testing the environment to ensure smooth training and testing processes, \
> (3) configuring server settings for diverse game modes, \
> (4) developing infrastructure for parallel training, and \
> (5) incorporating various other supporting features for research with the game, including quantitative metrics.
>
> In summary, we have constructed a comprehensive infrastructure that encompasses various controller, state, and action classes designed for multifaceted game control, thus facilitating interaction with the Freeciv server.
>
> Additionally, we have added support for tensor-based and LLM-based agents and have created a new game mode: "mini-game ([doc link](https://civrealm.github.io/civrealm/advanced_materials/minigame.html))". For more detailed documentation of our implementation, please refer to: <https://civrealm.github.io/civrealm/>.
>
> We sincerely appreciate your thoughtful review, and we hope we have addressed your concerns. Please do not hesitate to reach out if you require any further information.
>
> ---
> [ref1] Samvelyan, Mikayel, et al. "The starcraft multi-agent challenge.", 2019.

---

> > ### Author Response · Authors · 2023-11-22
> > **Seeking an open dialogue**
> >
> > Dear Reviewer,
> >
> > We sincerely appreciate the time and effort you have devoted to reviewing our work. We understand that your schedule may be quite busy. As the authors-reviewer discussion phase draws to a close, we kindly request your attention to our responses. Our aim is to gain insights into whether our responses effectively address your concerns and to ascertain if there are any additional questions or points you would like to discuss.
> >
> > We look forward to the opportunity for further discussion with you. Thank you for your thoughtful consideration.
> >
> > Best regards, The Authors

---

> ### Comment · Reviewer_XJ4N · 2023-11-22
>
> Thanks for your explanation! Hope you can understand that it is important for reviewer to have codes to review for these kinds of paper. I am happy to raise your score. :)
>
> Another question:
> In the table one, why Dota is not regarded as a stochastic game? I think the monster in the wild and the items drop from them are random. Why it is not multi-goal game? One may argue that the goal is single -- win the game, but I guess CivRealm is also to "win the game".

---

> > ### Author Response · Authors · 2023-11-23
> >
> > Thank you for your response! We truly appreciate your valuable input, which further enhanced the clarity of our paper.
> >
> > > In the table one, why Dota is not regarded as a stochastic game? I think the monster in the wild and the items drop from them are random.
> >
> > Thank you for pointing this out!  Upon further examination of the game mechanics, we have identified stochastic elements in Dota 2. We sincerely apologize for the oversight, and we have made the necessary updates to Table 1 to accurately reflect the stochastic nature of Dota.
> >
> > > Why is Dota not a multi-goal game? One may argue that the goal is single -- win the game, but I guess CivRealm is also to "win the game".
> >
> > Thank you for raising this interesting point. The victory condition of Dota is the destroy of the opponent's base, hence we classify Dota as a single-goal game. Conversely, CivRealm presents a range of diverse victory conditions. Players can achieve victory through domination (by conquering other players), scientific means (such as launching a spaceship), score-based criteria (aiming for the highest score), and so on. Each of these victory conditions demands a distinct and varied strategy, making CivRealm a multi-goal game.
> >
> > Furthermore, we have reevaluated the player count in Dota 2 and found that it remains constant throughout the gameplay. This is due to the fact that players are respawned after their characters are defeated. We have also updated Table 1 to reflect this. Once again, we greatly appreciate your feedback, which has contributed to the precision of our classification.

---

### Official Review · Reviewer_4Csd · 2023-10-31

**Soundness:** 3 good
**Presentation:** 2 fair
**Contribution:** 3 good
**Rating:** 8
**Confidence:** 4

**Summary:**

The paper introduces a novel testbed environment inspired by the Civilization game, called *CivRealm*, which is a multi-agent multi-goal long-horizon challenging environment. The authors also implement RL and LLM baseline agents and show that they struggle to make substantial progress in the full game.

**Strengths:**

1. The authors present a novel and challenging testbed for agent studies, which is a big contribution.
2. The experiments on the proposed agents also set up baselines that follow-up works can improve upon.
3. The environment also has mini-games that can benefit the (multi-agent) RL community.
4. The paper is clearly written and easy to follow.

**Weaknesses:**

1. Some details are not clear enough. As this is a multi-agent environment, what are the baseline RL and LLM agents play against? I didn't seem to find such details in the description.
2. The proposed baselines are not working in the (full game) environment. Maybe it is better to also set different difficulty levels (e.g. by different task horizons) for the full game so that later research can more easily be evaluated on the benchmark.
3.  Is Figure 2 the real situation that the LLM agents discover or the expected situation? If it is what the LLM agents discover, why does Tit still behave poorly?
4. The text in Figure 4 is too small to view.

**Questions:**

1. Why the success rate of the RL agent in some mini-games (e.g., SettlerBuildCity) is higher for the hard setting than for the easier settings?
2. Can the authors comment on the reason for instability and the sudden drop in performance in Figure 9?
3. It would be great if the authors could provide videos of the experiments or visualization ways in the future code repo.

---

> ### Author Response · Authors · 2023-11-18
> **Point-to-Point Response to Reviewer 4Csd**
>
> We thank reviewer 8arZ for the valuable time and constructive feedback.
>
> #### **Q1: Some details are not clear enough. As this is a multi-agent environment, what are the baseline RL and LLM agents play against? I didn't seem to find such details in the description.**
>
> **A1:**
> Thank you very much for bringing this to our attention. The opponent players that our baseline methods play against are built-in AIs implemented by the original Freeciv team. We have added this detail in Appendix A.1 of the revision.
>
> #### **Q2: The proposed baselines are not working in the (full game) environment. Maybe it is better to also set different difficulty levels (e.g. by different task horizons) for the full game so that later research can more easily be evaluated on the benchmark.**
>
> **A2:**
> Thank you for the suggestion! We will be adding APIs to allow users to configure the length of a full game. This will enable later research projects to evaluate themselves on full games with varying time horizons.
>
> The current full game is also highly customizable in several ways:
>
> - The number of opponents and the map size can be adjusted.
> - The level of resources on the map can be customized.
> - It offers different levels of difficulty in terms of the built-in AI opponents.
>
>
> #### **Q3: Is Figure 2 the real situation that the LLM agents discover or the expected situation? If it is what the LLM agents discover, why does Tit still behave poorly?**
>
> **A3:**
> Thank you for bringing this confusing part to our attention. This is an expected situation simulated by our built-in rule-based AI, which we utilize to illustrate the complexity of the Freeciv game. Figure 8 displays what the LLM agents have discovered.
>
> #### **Q4: The text in Figure 4 is too small to view.**
>
> **A4:**
> Thank you for the reminder. We have updated Figure 4 in the revised paper.
>
> #### **Q5: Why the success rate of the RL agent in some mini-games (e.g., SettlerBuildCity) is higher for the hard setting than for the easier settings?**
>
> **A5:**
> We have also observed this interesting phenomenon. This counterintuitive situation arises from a *shortcut strategy* discovered by the RL agent. Specifically, as outlined in Appendix A.2 of the paper and detailed at <https://civrealm.github.io/civrealm/advanced_materials/minigame.html>, the winning condition for the hard mode of the SettlerBuildCity task is determined by averaging the potential city positions on the minigame map. In the harder mode, the land terrain is considerably less favorable, with an abundance of deserts, tundra, and swamps, resulting in lower tile production values for food, production, and trade. Consequently, the winning score is lower due to the averaging across these poorer terrains, resulting in a ranking of minimal winning scores as follows: easy > normal > hard.
>
> Originally, the goal of the hard mode was to locate and settle in a resource-rich land area (e.g., forests or rivers) amidst these challenging terrains. However, sea and ocean tiles always offer more resources compared to the widely distributed poor land terrains. We have observed that tensor-based agents tend to exploit this unique scenario and learn a shortcut strategy of exclusively building coastal cities. This approach easily results in above-average production values for inland terrain, meeting the winning criteria for settlerBuildCity[hard].
>
> #### **Q6: Can the authors comment on the reason for instability and the sudden drop in performance in Figure 9?**
>
> **A6:**
> The sudden drop is attributed to the player encountering opponents or a pirate invasion, which could trigger when the player has reached a certain development stage. Our method, Mastaba, has outpaced the development of BaseLang; therefore, Mastaba encountered pirate invasions and was defeated at an earlier stage. It is worth noting that in the game, defending against pirate invasions is challenging, even for human players, as the pirates' technology, attack strength, and other factors are typically set to a high difficulty level.
>
> #### **Q7: It would be great if the authors could provide videos of the experiments or visualization ways in the future code repo.**
>
> **A7:**
> We included two videos in the supplementary materials: "Mastaba" and "BaseLang," each showcasing a complete game. In terms of visualization, our codebase offers users two functionalities: 1) the ability to visually observe the agent playing the game and 2) the option to automatically record screenshots of the game. We also customized the [FCIV-NET](https://github.com/fciv-net/fciv-net) version of the game for users, which provides a 3D visualization (shown in Fig 1).
>
>
> We appreciate again your thoughtful review and we hope we addressed your concerns. Please let us know if you'd like any further information.

---

> > ### Comment · Reviewer_4Csd · 2023-11-22
> > **Response to authors**
> >
> > I am satisfied with the authors' response. I will keep my original score.

---

> > > ### Author Response · Authors · 2023-11-22
> > > **Thank you for your response**
> > >
> > > Thank you for your response. We genuinely appreciate your thoughtful feedback, which further improves the clarity of our paper.
> > >
> > > Best,
> > >
> > > Authors

---

### Official Review · Reviewer_3D5Q · 2023-11-03

**Soundness:** 3 good
**Presentation:** 3 good
**Contribution:** 3 good
**Rating:** 8
**Confidence:** 3

**Summary:**

This is a dataset and benchmark paper that introduces the CivRealm environment. It supports a number of game features based on the Civilization game, including an agent-agnostic framework, a friendly interface, and a tensor-based agent interface, supporting a variety of custom tasks with custom goal definitions, etc. This is a very exhaustive benchmark for a long-horizon strategy-based game. The paper also discusses the performance of three methods (one tensor-based and one LLM-based).

**Strengths:**

1. This dataset and associated framework is very useful and will support a lot of research for multi-agent settings involving various kinds of agents.
2. The whole framework is very novel.

**Weaknesses:**

I'm not sure how robust the client-server architecture used in the framework is. There are no comments on the robustness of this.

**Questions:**

1. How robust is the server-client architecture? Did you do any experiments to test it?
2. Is there support for designing custom games? I see Lua script being mentioned to specify the reward structure, new goals, etc. Is there also a support for new maps?

---

> ### Author Response · Authors · 2023-11-18
> **Point-to-Point Response to Reviewer 3D5Q**
>
> We thank reviewer 3D5Q for the valuable time and constructive feedback.
>
> #### **Q1: How robust is the server-client architecture? Did you do any experiments to test it?**
>
> **A1:**
> That is an excellent question! During the development of CivRealm, we adhered to the test-driven development practice. We created unit tests, integration tests, and end-to-end tests to verify the reliability of various game operations and the training environment. Additionally, we conducted testing on multiple platforms, including Linux, Mac OS, and Windows, to ensure smooth gameplay for our agents to the best of our efforts. Most of the test code is available in the [civrealm/tests](https://github.com/civrealm/civrealm/tree/main/tests) folder within the released `civrealm` repository for your reference.
>
> #### **Q2: Is there support for designing custom games? I see Lua script being mentioned to specify the reward structure, new goals, etc. Is there also a support for new maps?**
>
> **A2:**
> Yes, we have uploaded a `civrealm-sav` repository in our released code. This repository enables users to customize new games and generate new maps. For a detailed tutorial, please refer to the 'Create new Mini-Game' content at: <https://civrealm.github.io/civrealm/advanced_materials/minigame.html>.
>
> We sincerely appreciate your thoughtful review, and we hope we have addressed your concerns. Please let us know if you would like any further information.

---

> > ### Comment · Reviewer_3D5Q · 2023-11-20
> > **Thanks for clarification**
> >
> > I read the rebuttal carefully and I thank the authors for their response and their work.

---

> ### Author Response · Authors · 2023-11-21
> **Thank you for your response**
>
> Thank you for your prompt response. We are genuinely grateful for your thoughtful feedback, which further improves the clarity of our paper.
>
> Best,
>
> Authors

---

### Public Comment · ~Guohao_Li1 · 2023-11-14
**Suggesting related work**

This work presents an environment inspired by the Civilization game, designed to challenge decision-making agents in learning and reasoning. It introduces CivRealm, a multi-agent, imperfect-information, general-sum game with dynamic player numbers for cooperative, competitive or mix settings. This environment requires agents to deal with stochastic settings demanding diplomacy and negotiation skills. Two agent types are explored: tensor-based (focusing on learning) and LLM-based (emphasizing reasoning). Preliminary results show reasonable performance in mini-games, but struggles in the full game, highlighting the complexity and novelty of CivRealm.

Thanks for the awesome work. It could also be beneficial to discuss prior work on multi-LLM agents for the study of cooperative settings [1].

[1] Li, Guohao, Hasan Abed Al Kader Hammoud, Hani Itani, Dmitrii Khizbullin, and Bernard Ghanem. "CAMEL: Communicative Agents for" Mind" Exploration of Large Language Model Society." NeurIPS 2023

---

> ### Author Response · Authors · 2023-11-18
>
> Thank you for your insightful feedback on our work, and we appreciate your positive comments.
> We agree that discussing prior work on multi-LLM agents for the study of cooperative settings [1] would be a valuable addition to our paper. Your feedback is much appreciated.

---

### Author Response · Authors · 2023-11-18
**Summary of Revisions**

To all reviewers:

We would like to express our sincere gratitude for your meticulous review and the insightful comments you provided. In response to your feedback:

1. We have released our code on GitHub at an anonymous account for review purposes: <https://github.com/civrealm>. and you can also access our documentation at: <https://civrealm.github.io/civrealm/>. This GitHub account hosts four distinct repositories <https://github.com/civrealm?tab=repositories>:
   - `civrealm`: this repository contains the core code of the CivRealm environment, encompassing both tensor-based and language-based APIs, along with comprehensive documentation.
   - `civrealm-sav`: this repository contains the code responsible for creating the mini-games, which also enables users to customize their own games and generate new maps.
   - `civrealm-tensor-baseline`: this repository provides the code for our baseline tensor-based RL agents.
   - `civrealm-llm-baseline`: this repository hosts the code for our LLM agents, including BaseLang and Mastaba. Please note that this anonymous repository will transition to our official repository following the completion of the review process. The documentation will be continuously updated for more details.

2. We have incorporated a few updates into the paper as per the request of reviewer 4Csd. These updates include enlarging the font size of Figure 4 and providing additional information about the environment. To further improve the clarity, we also add more details about the infrastructure in Appendix and indicate that the environment is based on the Civilization game in the title.

We sincerely appreciate your time and effort in reviewing our work.

Sincerely,

The Authors

---

### Meta-Review · Area_Chair_tKGY · 2023-12-01

**Metareview:**

This paper contributes a new environment and benchmark for multi-agent systems, which can be used both with a tensor-based and language-based interface. All reviewers were in agreement that it is a useful contribution to the field.

There is one minor thing I noticed when reading the paper, which it would be nice to correct. The table comparing platforms claims that for Melting Pot the number of players does not increase or decrease during a game. That's not true though since the number players changes over time in quite a lot of Melting Pot environments.

**Justification For Why Not Higher Score:**

All reviewers agreed it should be accepted, though none said anything to suggest it should be an oral either.

**Justification For Why Not Lower Score:**

All reviewers agreed this paper should be accepted.

---

### Decision · Program_Chairs · 2024-01-16

Accept (spotlight)